# Extracellular matrix-inducing Sox9 promotes both basal progenitor proliferation and gliogenesis in developing neocortex

**Ayse Güven[1], Nereo Kalebic[1,2], Katherine R Long[1†], Marta Florio[1‡], Samir Vaid[1], Holger Brandl[1], Denise Stenzel[1§], Wieland B Huttner[1]***

[1]Max Planck Institute of Molecular Cell Biology and Genetics, Dresden, Germany; [2]Human Technopole, Milan, Italy

**Abstract** Neocortex expansion is largely based on the proliferative capacity of basal progenitors (BPs), which is increased by extracellular matrix (ECM) components via integrin signaling. Here we show that the transcription factor Sox9 drives expression of ECM components and that laminin 211 increases BP proliferation in embryonic mouse neocortex. We show that Sox9 is expressed in human and ferret BPs and is required for BP proliferation in embryonic ferret neocortex. Conditional Sox9 expression in the mouse BP lineage, where it normally is not expressed, increases BP proliferation, reduces Tbr2 levels and induces Olig2 expression, indicative of premature gliogenesis. Conditional Sox9 expression also results in cell-non-autonomous stimulation of BP proliferation followed by increased upper-layer neuron production. Our findings demonstrate that Sox9 exerts concerted effects on transcription, BP proliferation, neuron production, and neurogenic vs. gliogenic BP cell fate, suggesting that Sox9 may have contributed to promote neocortical expansion.

***For correspondence:**
huttner@mpi-cbg.de

[§]Present name: Denise Dreiseidler

**Present address:** [†]Centre for Developmental Neurobiology, MRC Centre for Neurodevelopmental Disorders, King's College London, London, United Kingdom; [‡]Harvard Medical School, Department of Genetics, Boston, United States

**Competing interests:** The authors declare that no competing interests exist.

## Introduction

The expansion of the neocortex in the course of human evolution and its growth during the development of the human brain have become a central topic of molecularly and cellularly focused developmental neuroscience. Neocortical expansion is thought to constitute one of the bases for the unique cognitive abilities of humans. At the cellular level, the proliferative capacity and pool size of cortical neural progenitor cells (cNPCs) is regarded as a key parameter underlying neocortical expansion (*Borrell and Reillo, 2012*; *Dehay et al., 2015*; *Fernández et al., 2016*; *Fietz and Huttner, 2011*; *Fish et al., 2008*; *Florio and Huttner, 2014*; *Kriegstein et al., 2006*; *Lui et al., 2011*; *Martynoga et al., 2012*; *Miller et al., 2019*; *Mitchell and Silver, 2018*; *Molnár et al., 2019*; *Namba and Huttner, 2017*; *Rakic, 2009*; *Silbereis et al., 2016*; *Uzquiano et al., 2018*; *Velasco et al., 2019*).

There are two major germinal zones in the developing neocortex, which harbor two principal classes of cNPCs that exhibit distinct features related to the apical-basal polarity of the cortical wall. The apical-most, and primary, germinal zone, the ventricular zone (VZ), harbors the cell bodies of cNPCs collectively referred to as apical progenitors, all of which contact the ventricle (*Taverna et al., 2014*). Among these, apical (or ventricular) radial glia (aRG), which arise by transition from neuroepithelial cells as the generation of neocortical neurons begins (*Götz and Huttner, 2005*; *Kriegstein and Götz, 2003*) and which exhibit pronounced apical-basal cell polarity (*Götz and Huttner, 2005*; *Reiner et al., 2012*; *Taverna et al., 2014*), have been recognized as a major cNPC type (*Lui et al., 2011*; *Namba and Huttner, 2017*). The zone basal to the VZ, the subventricular zone

(SVZ), constitutes a secondary germinal layer that harbors the cell bodies of cNPCs collectively referred to as basal progenitors (BPs), all of which lack contact with the ventricle (*Borrell and Reillo, 2012*; *Dehay et al., 2015*; *Fietz and Huttner, 2011*; *Kriegstein and Alvarez-Buylla, 2009*; *Pontious et al., 2008*; *Taverna et al., 2014*). There are two main types of BPs, (i) basal intermediate progenitors (bIPs), which lack apical-basal cell polarity and do not exhibit significant cell processes at mitosis (*Attardo et al., 2008*; *Haubensak et al., 2004*; *Miyata et al., 2004*; *Noctor et al., 2004*), and (ii) basal (or outer) radial glia (bRG), which exhibit basal and/or apical cell polarity, extending one or more basal and/or apically directed cell processes throughout their cell cycle including mitosis (*Betizeau et al., 2013*; *Fietz et al., 2010*; *Hansen et al., 2010*; *Kalebic et al., 2019*; *Pilz et al., 2013*; *Reillo et al., 2011*; *Shitamukai et al., 2011*; *Wang et al., 2011*).

For cell biological reasons related to their apical cell polarity, aRG mitoses are confined to the ventricular surface, a limited space, which poses a constraint with regard to maximizing their number and, consequently, to increasing aRG pool size (*Fietz and Huttner, 2011*; *Fish et al., 2008*; *Taverna et al., 2014*). In contrast, this constraint does not exist for BPs. By virtue of these cells having delaminated from the ventricular surface, BPs have an intrinsic advantage compared to aRG with regard to maximizing the number of their mitoses and hence to increasing their pool size, as they can undergo mitosis virtually anywhere along the radial axis of the SVZ (*Betizeau et al., 2013*; *Fietz and Huttner, 2011*; *Fietz et al., 2010*; *Fish et al., 2008*; *Florio and Huttner, 2014*; *Hansen et al., 2010*; *Reillo et al., 2011*). Accordingly, neocortical expansion is thought to be linked to an increase in the proliferative capacity of BPs, resulting in their increased pool size and a thickening of the SVZ (*Borrell and Reillo, 2012*; *Dehay et al., 2015*; *Fernández et al., 2016*; *Fietz and Huttner, 2011*; *Florio and Huttner, 2014*; *Kriegstein et al., 2006*; *Lui et al., 2011*). Thus, in mammals lacking neocortical expansion, such as mouse, BPs exhibit only low proliferative capacity, typically dividing only once to generate two post-mitotic neurons, and their pool size is relatively small and the SVZ comparably thin (*Arai et al., 2011*; *Florio and Huttner, 2014*; *Haubensak et al., 2004*; *Miyata et al., 2004*; *Noctor et al., 2004*). In contrast, in mammals showing neocortical expansion, notably human, BPs exhibit high proliferative capacity, resulting in an increased pool size and a thick SVZ (*Betizeau et al., 2013*; *Fernández et al., 2016*; *Fietz et al., 2010*; *Florio and Huttner, 2014*; *Hansen et al., 2010*; *Pilz et al., 2013*; *Reillo et al., 2011*).

Hence, the crucial question is: what underlies the differences in BP proliferative capacity across the various mammals? An important clue pointing to the differential expression of extracellular matrix (ECM) components as a key regulatory parameter has come from comparative analyses of the transcriptomes of mouse vs. human VZ and SVZ as well as specific cNPC subpopulations. Specifically, in embryonic mouse neocortex, BPs down-regulate the endogenous expression of ECM components in comparison with aRG, in line with the lower proliferative capacity of the former than the latter (*Arai et al., 2011*). Consistent with this, the expression of ECM components is down-regulated in the embryonic mouse SVZ compared to the VZ, whereas in fetal human neocortex this expression is maintained not only in the VZ, but also inner SVZ (ISVZ) and outer SVZ (OSVZ) (*Fietz et al., 2012*). Accordingly, not only human aRG, but also human bRG show a characteristic expression of ECM components (*Florio et al., 2015*; *Pollen et al., 2015*). Moreover, when mimicking the physiological situation in the fetal human SVZ in an embryonic mouse model, the targeted activation of integrins, the canonical receptors for ECM components, specifically of integrin αvβ3, on mouse BPs promotes their proliferation (*Stenzel et al., 2014*). Conversely, inhibition of integrin αvβ3 in an *ex vivo* model of the developing neocortex of the ferret, which exhibits an expanded, folded cerebral cortex, reduces the BP pool size, in particular that of bRG (*Fietz et al., 2010*).

Furthermore, we have recently shown that BPs in species with an expanded neocortex, such as human and ferret, exhibit an increased number of cell processes; these are used to receive extrinsic pro-proliferative signals via integrin signaling, notably involving integrin β1 (*Kalebic et al., 2019*). Blocking of integrin β1 in fetal human neocortex thus resulted in a reduction of BP proliferation (*Kalebic et al., 2019*). Taken together, these findings have led to the general concept that an increased expression of ECM components by BPs contributes to generate a proliferative niche away from the ventricular surface and to promote BP proliferation via increased integrin signaling (*Fietz et al., 2010*; *Fietz et al., 2012*; *Kalebic et al., 2019*; *Long and Huttner, 2019*; *Stenzel et al., 2014*).

This in turn leads to the key question: which transcriptional machinery governs the differential expression of ECM components in the neocortical SVZ of the various mammals? An *in silico* analysis

of transcription factors expressed in embryonic mouse vs. fetal human germinal zones and predicted to bind to promoters of ECM genes has revealed Sox9 as a promising candidate to drive expression of ECM components in the human SVZ (*Fietz et al., 2012*). Here, we have studied the physiological expression of Sox9 in the VZ vs. SVZ of developing mouse, ferret and human neocortex and, based on the results obtained, examined the effects of conditional Sox9 expression in mouse BPs on their proliferative capacity and on driving the expression of ECM genes.

In this context, given the role of Sox9 in driving glia-specific gene expression (*Huang et al., 2015*; *Kang et al., 2012*; *Klum et al., 2018*; *Martini et al., 2013*; *Molofsky et al., 2013*; *Nagao et al., 2016*; *Selvaraj et al., 2017*) and in the neurogenesis-to-gliogenesis switch in the developing spinal cord (*Finzsch et al., 2008*; *Molofsky et al., 2013*; *Stolt et al., 2003*; *Stolt and Wegner, 2010*; *Wegner and Stolt, 2005*), a related issue concerns the neuronal vs. glial fate diversity of neocortical SVZ progenitors across species. In the developing mouse neocortex, neurogenesis and gliogenesis take place sequentially, that is, after neurogenesis, SVZ progenitors of oligodendrocytes and astrocytes are being generated (*Kriegstein and Alvarez-Buylla, 2009*; *Merkle et al., 2004*). In contrast, in the SVZ of gyrencephalic species, such as human, macaque and ferret, neurogenic and gliogenic progenitors co-exist at later stages of neurogenesis (*Martínez-Cerdeño et al., 2012*; *Rash et al., 2019*; *Reillo and Borrell, 2012*; *Reillo et al., 2011*; *Zecevic et al., 2005*). We therefore have also examined the effects of conditional Sox9 expression in mouse BPs on their neurogenic vs. gliogenic fate. Taken together, our data provide novel insight into the cell-autonomous vs. cell non-autonomous stimulation of BP proliferation, the consequences for neuron production, and the relationship between neurogenesis and gliogenesis.

## Results

### Sox9-expressing BPs occur in the SVZ of embryonic ferret and fetal human, but not embryonic mouse, neocortex

To gain an initial insight into Sox9 expression in developing neocortex, we analyzed the FPKM values for *Sox9/SOX9* mRNA in the germinal zones and in specific cNPC types of mouse and human neocortex at mid-neurogenesis, using two published transcriptome datasets (*Fietz et al., 2012*; *Florio et al., 2015*). In the germinal zones of embryonic day (E) 14.5 mouse neocortex, *Sox9* mRNA was found to be expressed in the VZ, but not SVZ (*Figure 1—figure supplement 1A*, left), whereas in the 13–16 weeks post conception (wpc) human neocortex, *SOX9* mRNA was found to be expressed not only in the VZ, but also in the ISVZ and OSVZ (*Figure 1—figure supplement 1A*, right) (*Fietz et al., 2012*). In both species, no significant *Sox9/SOX9* mRNA expression was observed in the cortical plate (CP). In specific cNPC types isolated from E14.5 mouse neocortex, *Sox9* mRNA was highly expressed in aRG, but not in bRG, bIPs and neurons (*Figure 1—figure supplement 1B*, left) (*Florio et al., 2015*). Within mouse aRG, *Sox9* mRNA levels were almost three times as high in the proliferative aRG subpopulation that lacks *Tis21*-GFP expression than in the neurogenic aRG subpopulation that exhibits *Tis21*-GFP expression. In contrast to mouse, in specific cNPC types isolated from 13 wpc human neocortex, *SOX9* mRNA was highly expressed in aRG, but was also found in bRG (*Figure 1—figure supplement 1B*, right) (*Florio et al., 2015*).

In light of these mRNA data, we analyzed the expression of the Sox9 protein in developing mouse and human neocortex by immunofluorescence. For these analyses, we included a third species, the ferret, which like human but in contrast to mouse exhibits an expanded SVZ containing BPs with a high proliferative capacity (*Borrell and Reillo, 2012*; *Fietz et al., 2010*; *Kalebic et al., 2018*; *Kawasaki, 2018*; *Reillo et al., 2011*; *Smart and McSherry, 1986*; *Turrero García et al., 2016*). Consistent with the *Sox9* mRNA data (see *Figure 1—figure supplement 1*), the Sox9 protein was restricted to the VZ in E14.5 mouse neocortex (*Figure 1A,G*), but was abundantly expressed in the VZ, ISVZ and OSVZ of E40 ferret (a typical developmental stage for the analysis of cNPCs in the context of cortical neurogenesis [*Kalebic et al., 2018*]) and 15 wpc human neocortex (*Figure 1B,C,H,I*). At the subcellular level, the Sox9 protein showed a nuclear localization. These data suggested that the Sox9 protein is expressed in BPs of ferret and human, but not mouse, neocortex.

In order to further explore the identity of the Sox9-expressing cells in the SVZ of developing ferret and human neocortex, we investigated a potential co-expression of Sox9 with Sox2 (*Pevny and Nicolis, 2010*), the expression of which among BPs is characteristic of proliferative BPs, notably bRG

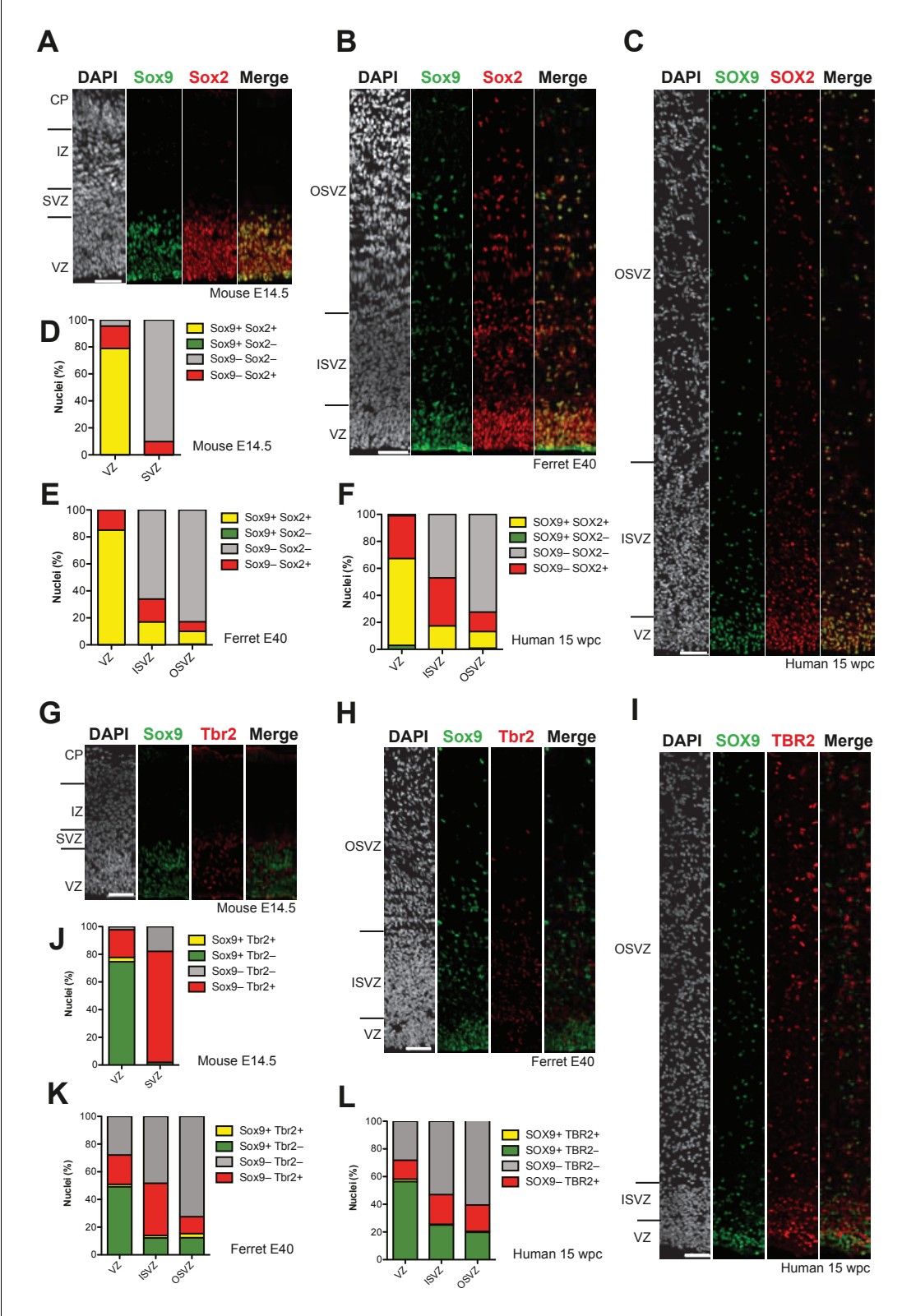

**Figure 1.** Sox9-expressing BPs occur in the SVZ of embryonic ferret and fetal human but not embryonic mouse neocortex. (A–C) Double immunofluorescence for Sox9 (green) and Sox2 (red), combined with DAPI staining (white), of mouse E14.5 (A), ferret E40 (B) and human 15 wpc (C) neocortex. (D–F) Quantification of the percentage of nuclei (identified by DAPI staining) that are Sox9 plus Sox2 double-positive (yellow), Sox9-positive only (green), Sox2-positive only (red), and Sox9 plus Sox2 double-negative (gray), in mouse E14.5 (D), ferret E40 (E) and human 15 wpc (F) neocortex.
*Figure 1 continued on next page*

*Figure 1 continued*

(G–I) Double immunofluorescence for Sox9 (green) and Tbr2 (red), combined with DAPI staining (white), of mouse E14.5 (G), ferret E40 (H) and human 15 wpc (I) neocortex. (J–L) Quantification of the percentage of nuclei (identified by DAPI staining) that are Sox9 plus Tbr2 double-positive (yellow), Sox9-positive only (green), Tbr2-positive only (red), and Sox9 plus Tbr2 double-negative (gray), in mouse E14.5 (J), ferret E40 (K) and human 15 wpc (L) neocortex. (A–C, G–I) Ventricular surface is down. Upper margins of images in (A, B, G, H) correspond to the pial surface (A, G) and the basal boundary of the OSVZ (B, H); in (C, I) most but not all of the OSVZ is shown due to space constraints. Scale bars, 50 µm.

The online version of this article includes the following figure supplement(s) for figure 1:

**Figure supplement 1.** *Sox9* mRNA is expressed in fetal human, but not embryonic mouse, BPs.

(*Graham et al., 2003*; *Hansen et al., 2010*; *Reillo et al., 2011*; *Wang et al., 2011*). Double immunofluorescence showed that virtually all Sox9-expressing cells in the ISVZ and OSVZ of E40 ferret and 15 wpc human neocortex co-expressed Sox2, implying that these cells were proliferative BPs, and that these cells comprised about half of the Sox2-positive BPs (*Figure 1B,C,E,F*). Of note, regarding the VZ of developing mouse, ferret and human neocortex, while nearly all cells were Sox2-positive, 70–80% of these also expressed Sox9 (*Figure 1A–F*), again in line with these cells being cNPCs.

Analysis of expression of the transcription factor Tbr2 (encoded by *Eomes*) allows one to identify newborn bIPs in the VZ and can aid in distinguishing between neurogenic and proliferative BPs in the SVZ (*Englund et al., 2005*; *Kalebic et al., 2018*; *Kowalczyk et al., 2009*; *Pontious et al., 2008*; *Sessa et al., 2008*). Double immunofluorescence of the E14.5 mouse, E40 ferret and 15 wpc human neocortex for Sox9 and Tbr2 revealed that in all germinal zones, virtually none of the Sox9-positive cells expressed Tbr2, indicative of a mutually exclusive expression pattern (*Figure 1G–L*). Regarding the VZ, these data imply that the Sox9 and Sox2 double-positive cells observed in the three species (*Figure 1D–F*) are aRG, whereas the Sox9-negative but Sox2-positive cells (*Figure 1D–F*) are newborn BPs, notably newborn bIPs. Regarding the SVZ, these data suggest that the Sox9 and Sox2 double-positive cells observed in ferret and human are proliferative BPs, likely bRG, rather than neurogenic bIPs. The latter conclusion in turn implies that Sox9 expression is down-regulated upon BPs becoming committed to neurogenesis.

## Sox9-expressing BPs in ferret and human can re-enter the cell cycle and include bRG

The data described so far prompted us to further study the Sox9-expressing cNPCs, notably the BPs, in a gyrencephalic cortex and to analyze their proliferative capacity. For this purpose, we used ferret kits, whose neurogenesis still continues in the first five postnatal days, allowing application of a broad spectrum of experimental approaches (*Fietz et al., 2010*; *Gertz et al., 2014*; *Kawasaki, 2018*; *Reillo and Borrell, 2012*; *Turrero García et al., 2016*). We first performed immunofluorescence for the cell cycle marker PCNA on postnatal day (P) two and P3 ferret neocortex and found that the vast majority of the Sox9-positive cells in the three germinal zones were cycling (*Figure 2A, A′, B*).

We next determined if the Sox9-positive cNPCs were capable of cell cycle re-entry. To this end, we administered the thymidine analog EdU to P0 ferrets in order to label cells in S-phase, and sacrificed the kits at P2 and P3, that is, after a time interval that – in light of the known cell cycle length of ferret postnatal cNPCs (*Turrero García et al., 2016*) – should allow the labeled cells to go through mitosis and the resulting daughter cells to become either post-mitotic or to re-enter the cell cycle. Cell cycle re-entry was assessed by immunofluorescence for PCNA of Sox9 and EdU double-positive cells (*Figure 2A and A′*). We observed that the vast majority of the Sox9-positive progeny in the VZ and OSVZ can re-enter the next cell cycle at P2/P3 (*Figure 2C*), suggesting that the daughter cells of the EdU-labeled cNPCs maintained proliferative capacity. We conclude that in the developing ferret neocortex, the Sox9-expressing BPs in the OSVZ, like the Sox9-expressing aRG in the VZ, exhibit a high proliferative capacity.

We sought to extend this investigation to fetal human neocortex and to focus specifically on bRG, which constitute a major BP subpopulation in the human SVZ (*Fietz et al., 2010*; *Hansen et al., 2010*). To identify Sox9-expressing bRG, we combined immunofluorescence for Sox9 with that for phospho-vimentin, which labels cNPCs in mitosis and – by staining the major radial processes – marks mitotic bRG (*Fietz et al., 2010*; *Florio et al., 2015*; *Hansen et al., 2010*; *Figure 2D*).

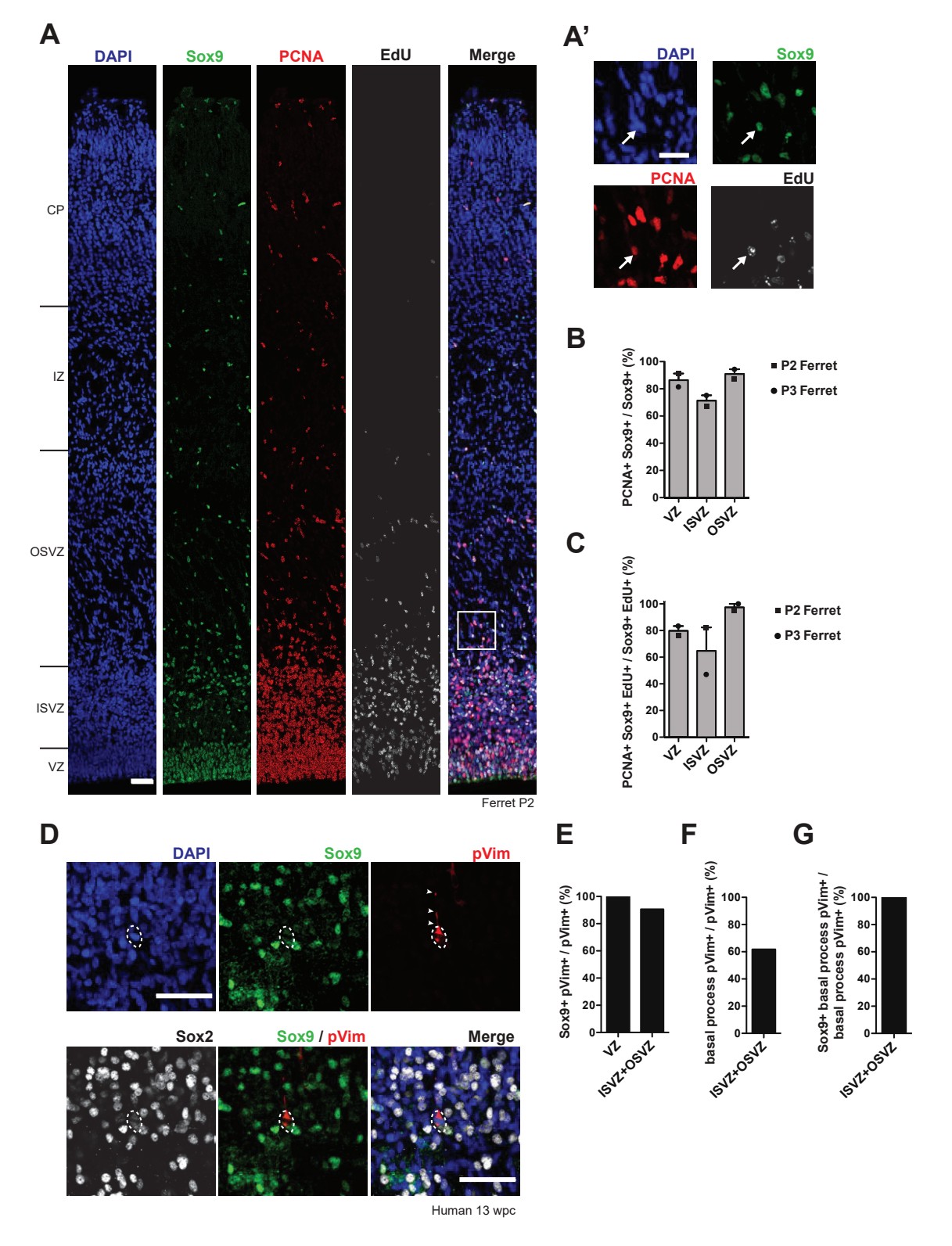

**Figure 2.** Sox9-expressing BPs in the SVZ of developing ferret and human neocortex are capable of cell cycle re-entry and include bRG. (**A**) Triple (immuno)fluorescence for Sox9 (green), PCNA (red) and EdU (white), combined with DAPI staining (blue), of P2 ferret neocortex. EdU was administrated at P0. Boxed area in (**A**) is shown in higher magnification in (**A'**). White arrows indicate a nucleus that is triple-positive for Sox9, PCNA and EdU, that is a Sox9-positive cycling BP in the OSVZ. (**B**) Quantification of the percentage of Sox9-positive nuclei that are PCNA-positive, that is the percentage of

*Figure 2 continued on next page*

Figure 2 continued

Sox9-positive cells that are cycling, in the indicated germinal zones of P2 (squares) and P3 (circles) ferret neocortex. Bars indicate the range between the individual values at P2 and P3. (C) Quantification of the percentage of Sox9 and EdU double-positive nuclei that are PCNA-positive, that is the percentage of Sox9-positive cNPCs that have re-entered the cell cycle, in the indicated germinal zones of P2 (squares) and P3 (circles) ferret neocortex. EdU was administered at P0. Bars indicate the range between the individual values at P2 and P3. (D) Triple immunofluorescence for Sox9 (green), phospho-vimentin (pVim, red) and Sox2 (white), combined with DAPI staining (blue), in the OSVZ of 13 wpc human neocortex. Arrowheads indicate the basal process, and dashed lines delineate the mitotic cell body, of a bRG. (E) Quantification of the percentage of pVim-positive cells, that is of mitotic cNPCs, that are Sox9-positive, in the indicated germinal zones of 13 wpc human neocortex. (F) Quantification of the percentage of pVim-positive cells that bear a basal process, that is the percentage of mitotic BPs that are bRG, in the ISVZ plus OSVZ of 13 wpc human neocortex. (G) Quantification of the percentage of basal process-bearing pVim-positive cells, that is of mitotic bRG, that are Sox9-positive, in the ISVZ plus OSVZ of 13 wpc human neocortex. (A, A', D) Scale bars, 50 μm (A, D) and 25 μm (A'). In (A), ventricular surface is down.

In 13 wpc human neocortex, essentially all mitotic aRG in the VZ and the overwhelming majority of the mitotic BPs in the SVZ were Sox9-positive (*Figure 2E*). Moreover, in line with previous observations (*Fietz et al., 2010*), the mitotic BPs exhibited a basal process and thus were bRG (*Figure 2F*). Importantly, all of the latter expressed Sox9, indicating that bRG in the fetal human neocortex are Sox9-positive (*Figure 2G*).

## CRISPR/Cas9-mediated knockout of Sox9 in embryonic ferret neocortex reduces BP proliferation

To examine if Sox9 is actually required for BP proliferation in a gyrencephalic cortex, we sought to genetically ablate it in embryonic ferret neocortex at E33, when the proliferation rates of the ferret BPs are high and ferret BPs exhibit all the morphological heterogeneity found in the human BPs at mid-neurogenesis (*Kalebic et al., 2019*). To this end, we established a genome editing approach in embryonic ferret neocortex that we previously developed in the embryonic mouse neocortex, which consists of *in utero* electroporation of recombinant Cas9 protein in a complex with guide RNAs (gRNAs) (*Kalebic et al., 2016*). Direct delivery of the recombinant Cas9/gRNA complex enables fast genome editing within the same cell cycle and reduces off-target effects (*Kalebic et al., 2016*). We targeted the *Sox9* locus by two gRNAs that are both complementary to the sequences in the first protein-coding exon (*Figure 3—figure supplement 1A*). We examined the efficiency of the knockout (KO) four days after *in utero* electroporation, at E37, and detected a strong reduction of the Sox9 protein in all germinal zones of the ferret neocortex (*Figure 3—figure supplement 1B,B', C*).

We examined the effects of Sox9 KO on BP proliferation by immunofluorescence for PCNA, a marker of cycling cells (*Figure 3A*), and found a striking reduction in proportion of PCNA+ cells throughout the SVZ, and particularly in the OSVZ (50% reduction) (*Figure 3B*). Immunofluorescence of pVim (*Figure 3—figure supplement 1B*), a marker of mitotic cells, corroborated these data, showing a decrease in mitoses throughout the ferret germinal zones (*Figure 3C*).

To assess the effects of Sox9 on proliferative BPs, we performed immunofluorescence for Sox2 (*Figure 3—figure supplement 2*) and found a remarkable 70% reduction in Sox2-positive BPs upon Sox9 KO (*Figure 3D*). Finally, to examine if these effects pertained also to bRG, the subset of BPs that is thought to be instrumental for the evolutionary expansion of the neocortex, we focused on Sox2-positive cells in the SVZ that exhibited a radial orientation of the nucleus and the presence of radial processes (*Kalebic et al., 2018*) (basal and/or apical, see *Figure 3—figure supplement 2*, inset for examples of a bRG and a multipolar BP). We found a decrease in the proportion of these cells in the total SVZ, and in particular in the OSVZ, upon Sox9 KO (*Figure 3E*).

Taken together, our data show that CRISPR/Cas9-mediated KO of Sox9 in embryonic ferret neocortex results in a decrease of BP proliferation. Importantly, this decrease pertains to both types of BPs (multipolar BPs and bRG).

## Conditional Sox9 expression in mouse BPs increases their proliferation and cell cycle re-entry

The absence of Sox9 expression in mouse BPs (*Figure 1*), which are known to have a low proliferative capacity, vs. its presence in the highly proliferative human BPs (*Figure 2*) and its requirement for the abundance of the highly proliferative ferret BPs (*Figure 3*), led us to hypothesize that Sox9 could promote BP proliferation. We therefore sought to conditionally express Sox9 in mouse BPs (in

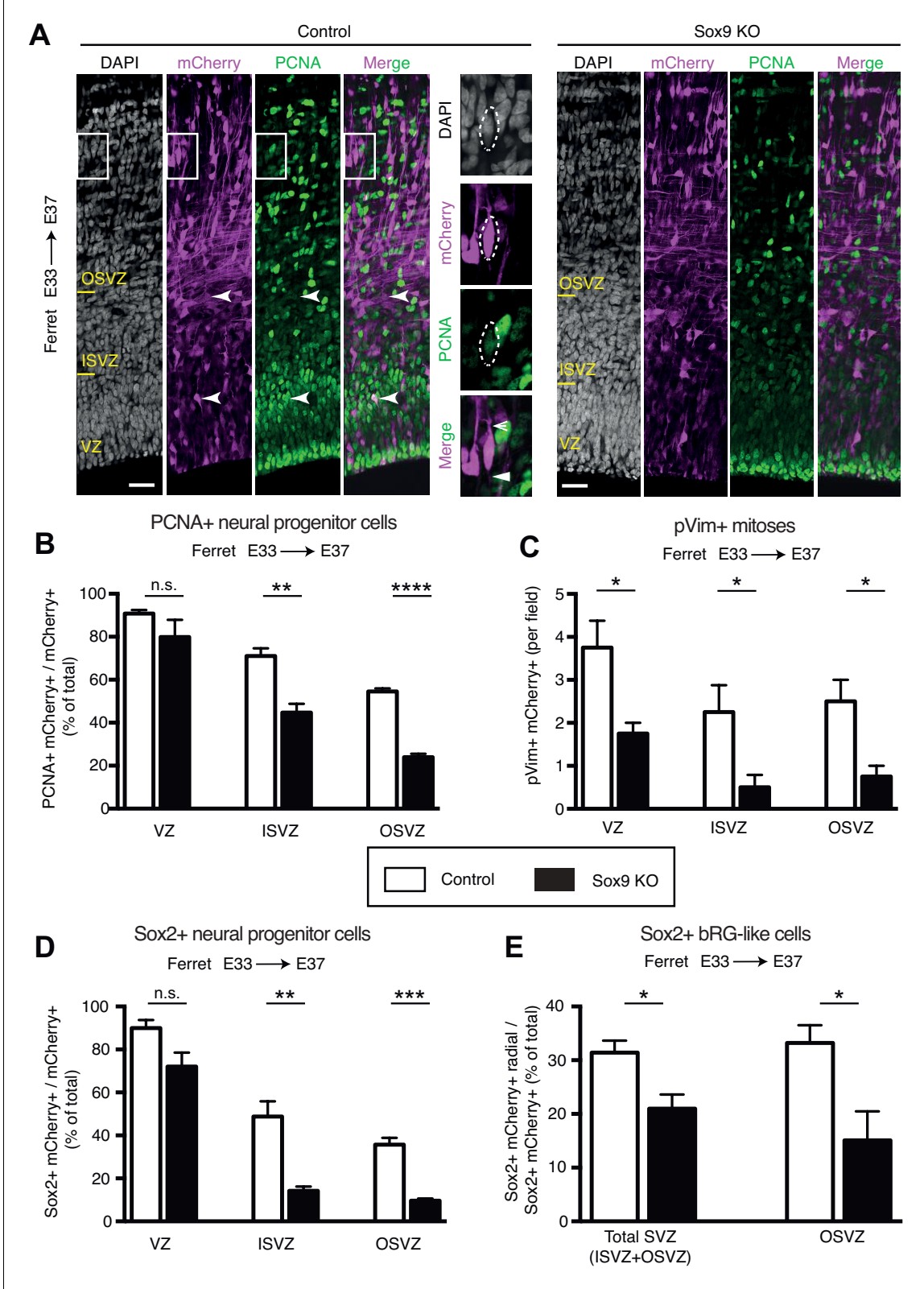

**Figure 3.** CRISPR/Cas9-mediated KO of *Sox9* in embryonic ferret neocortex leads to a reduction in BP proliferation. Ferret E33 neocortex was electroporated *in utero* with a plasmid encoding mCherry together with a complex of recombinant Cas9 protein with gRNAs targeting either Sox9 (Sox9 KO) or LacZ (Control), followed by analysis at E37. (**A**) Double immunofluorescence for mCherry (magenta) and PCNA (green), combined with DAPI staining (white). Images are single optical sections. Scale bars, 30 μm. Boxes (35 × 50 μm) indicate mCherry-positive bRG in the OSVZ, shown at
*Figure 3 continued on next page*

*Figure 3 continued*

higher magnification on the right. Insets: dashed lines, cell body contour; arrow, a bifurcated basal process; arrowhead, an apically-directed process. (**B**) Percentage of mCherry-positive cells in the VZ, ISVZ and OSVZ that are PCNA-positive in control (white) and Sox9 KO (black). Note that in the two PCNA immunostainings shown in (**A**), the brightest cells are equally bright for both the control and the Sox9 KO, as is obvious when comparing (i) the PCNA+ cells at the apical surface, and (ii) the brightest cells in the OSVZ. In contrast, in the basal region of the VZ, the intensity of immunostaining of the PCNA-positive cells is less in the Sox9 KO than the control. Importantly, such differences in PCNA immunostaining intensity did not affect the quantification shown, because even only weakly PCNA-positive cells were scored as PCNA+. (**C**) Quantification of mCherry-positive mitotic cells in the VZ, ISVZ and OSVZ, as revealed by pVim immunofluorescence (see *Figure 3—figure supplement 1* for the immunofluorescence images), in a 100 µm-wide field of the cortical wall, in control (white) and Sox9 KO (black). (**D**) Percentage of mCherry-positive cells in the VZ, ISVZ and OSVZ that are Sox2-positive in control (white) and Sox9 KO (black) (see *Figure 3—figure supplement 2* for the immunofluorescence images). (**E**) Percentage of mCherry-positive cells in total SVZ and separately OSVZ that are Sox2-positive with a radial orientation of the nucleus and presence of at least one radial process (bRG-like cells) in control (white) and Sox9 KO (black). (**B–E**) Data are the mean of 4 embryos from three different litters. Error bars indicate SD; *, p<0.05; **, p<0.01; ***, p<0.001; ****, p<0.0001; n.s., not statistically significant; Student's t-test.

The online version of this article includes the following figure supplement(s) for figure 3:

**Figure supplement 1.** CRISPR/Cas9-mediated KO of *Sox9* in embryonic ferret neocortex ablates Sox9 expression *in vivo*.

**Figure supplement 2.** CRISPR/Cas9-mediated KO of *Sox9* in embryonic ferret neocortex reduces abundance of Sox2-positive BPs.

addition to its physiological expression in aRG), to see if this would lead to increased mouse BP proliferation. To this end, we generated a conditional Sox9 expression construct (*Figure 4A*) and introduced it by *in utero* electroporation into aRG of tamoxifen-treated E13.5 embryos of the *Tis21*-CreER^T2 mouse line, which allows expression of floxed constructs specifically in BP-genic aRG and the BP progeny derived therefrom (*Wong et al., 2015*; *Figure 4B*). When using the conditional Sox9 expression construct in conjunction with tamoxifen-induced Cre-mediated recombination, expression of nuclear RFP is indicative of Sox9 expression (*Figure 4A*).

We first validated the conditional Sox9 expression construct by transfecting it into HEK293T cells, which do not express endogenous Sox9 (*Figure 4—figure supplement 1A,B*), with or without co-transfection of a construct driving strong constitutive expression of nuclear Cre. Transfection of the conditional Sox9 expression construct alone led to the expression of GFP from a floxed cassette, but not of RFP, the expression of which depends on the removal of the floxed GFP cassette (*Figure 4—figure supplement 1C*, see also *Figure 4A*). In contrast, transfection of the conditional Sox9 expression construct in combination with the Cre-expressing construct resulted in the expression of RFP but not GFP, and all RFP-expressing cells were positive for Sox9 (*Figure 4—figure supplement 1D*). These observations provided a validation of the conditional Sox9 expression construct. It is worth noting that whereas in these transfected HEK293T cells the RFP was observed in the nucleoplasm, Sox9 immunoreactivity was also seen outside the nucleoplasm, likely reflecting the strong overexpression of Sox9 (*Figure 4—figure supplement 1D*).

Next, we validated the conditional expression of Sox9 in mouse BP-genic aRG and their BP progeny using the *Tis21*-CreER^T2 line. We induced translocation of Cre to the nucleus by tamoxifen administration at E12.5 and E13.5, introduced either the control RFP-expressing construct (see *Figure 4A*) or the conditional Sox9 expression construct into neocortical aRG by *in utero* electroporation at E13.5, and performed immunofluorescence analyses at E14.5 (see *Figure 4B*). Upon electroporation of the control construct, we observed expression of RFP predominantly in the SVZ, whereas the expression of the endogenous Sox9 was confined to the VZ (*Figure 4—figure supplement 2A*), consistent with the data described above (see *Figure 1A*). In contrast, upon electroporation of the conditional Sox9 expression construct, strong Sox9 immunoreactivity was observed not only in the VZ, but also in the SVZ (*Figure 4—figure supplement 2B*). Importantly, all of the strongly Sox9-positive, that is exogenous Sox9-expressing, cells co-expressed nuclear RFP (*Figure 4—figure supplement 2B*), demonstrating that RFP expression from the conditional Sox9 expression construct can be taken as an indicator of Sox9 expression. Furthermore, these data show that the use of the conditional Sox9 expression construct in tamoxifen-treated *Tis21*-CreER^T2 embryos is a means of eliciting Sox9 expression in mouse BPs.

We analyzed the distribution of the RFP-positive, that is Sox9-expressing, cells across the radial axis of the mouse cortical wall at E14.5 (*Figure 4—figure supplement 2C,D*) and E15.5 (*Figure 4—figure supplement 2E*). This did not reveal any significant differences in this distribution between control and conditional Sox9 expression, neither at E14.5 (*Figure 4—figure supplement 2D*) nor at

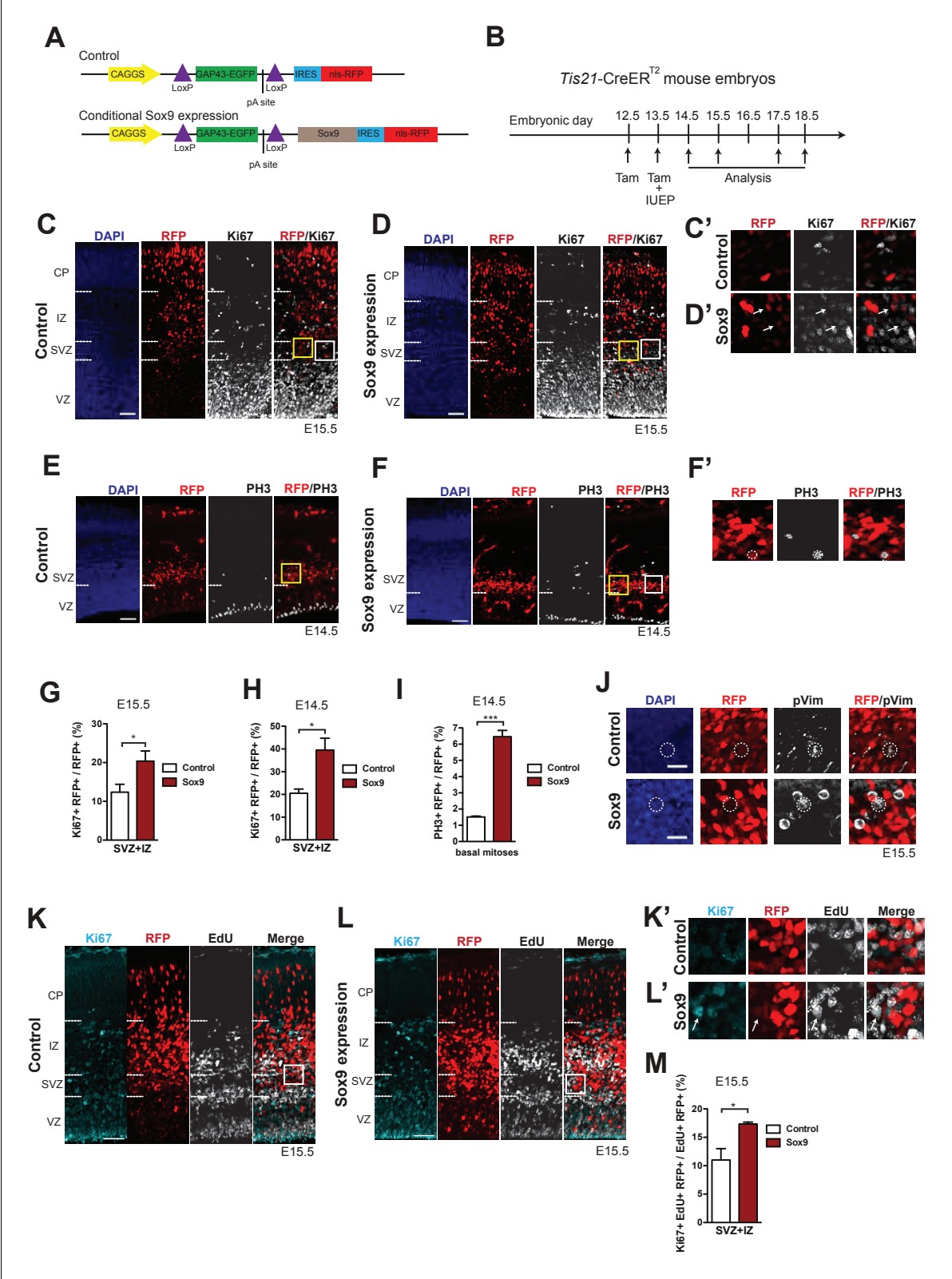

**Figure 4.** Conditional Sox9 expression in BPs of embryonic mouse neocortex increases their proliferation and cell cycle re-entry. (**A**) Constructs used to conditionally express nuclear RFP without (top, control construct) and with (bottom, conditional Sox9 expression construct) Sox9 in mouse BPs and their progeny using the *Tis21*-CreER[T2] line (see **B**). (**B**) Workflow of tamoxifen administration (Tam) at E12.5 and E13.5, *in utero* electroporation (IUEP) at E13.5, and immunostaining analyses of the neocortex at the indicated time points (arrows) yielding the data shown in this figure and subsequent

*Figure 4 continued on next page*

Figure 4 continued

figures, using heterozygous *Tis21*-CreER^T2 mouse embryos. (C, D) Double immunofluorescence of neocortex for RFP (red) and Ki67 (white), combined with DAPI-staining (blue), 48 hr after electroporation of control construct (C) or conditional Sox9 expression construct (D). Dashed lines indicate the borders between VZ, SVZ, IZ and CP. White boxed areas of the SVZ in (C) and (D) are shown at higher magnification in (C') and (D'), respectively; arrows indicate RFP-positive nuclei that are Ki67-positive. Yellow boxed areas of the SVZ in (C) and (D) are shown at higher magnification in **Figure 7A**, top row and bottom row, respectively. (E, F) Double immunofluorescence of neocortex for RFP (red) and phosphohistone H3 (PH3, white), combined with DAPI staining (blue), 24 hr after electroporation of control construct (E) or conditional Sox9 expression construct (F). Dashed lines indicate the border between VZ and SVZ. White boxed area in (F) is shown at higher magnification in (F'); dashed circles delineate a basal, PH3 and RFP double-positive mitosis. Yellow boxed areas of the SVZ in (E) and (F) are shown at higher magnification in **Figure 7B**, top row and bottom row, respectively. (G, H) Quantifications in the neocortical SVZ plus IZ, upon electroporation of control construct (white columns) or conditional Sox9 expression construct (red columns). (G) Quantification of the percentage of RFP-positive nuclei that are Ki67-positive, 48 hr after electroporation. (H) Quantification of the percentage of RFP-positive nuclei that are Ki67-positive, 24 hr after electroporation. Related representative images are not shown. (I) Quantification of the percentage of abventricular RFP-positive cells in neocortex that undergo basal mitosis as revealed by PH3 immunofluorescence, 24 hr after electroporation of control construct (white column) or conditional Sox9 expression construct (red column). (J) Double immunofluorescence of neocortex for RFP (red) and phospho-vimentin (pVim, white), combined with DAPI staining (blue), of the SVZ, 48 hr after electroporation of control construct (white column) or conditional Sox9 expression construct (red column). Dashed circles delineate the cell body of RFP and pVim double-positive cells. (K, L) Triple (immuno)fluorescence of neocortex for Ki67 (cyan), RFP (red) and EdU (white) 48 hr after electroporation of control construct (K) or conditional Sox9 expression construct (L). A single pulse of EdU was administered at E14.5, that is 24 hr after electroporation and 24 hr prior to analysis. Dashed lines indicate the borders between VZ, SVZ, IZ and CP. White boxed areas of the SVZ in (K) and (L) are shown at higher magnification in (K') and (L'), respectively; arrows indicate RFP-positive nuclei that are Ki67- and EdU-positive. (M) Quantification of the percentage of RFP and EdU double-positive nuclei in the neocortical SVZ plus IZ that are Ki67-positive, that is the percentage of RFP+ BPs that have re-entered the cell-cycle, 48 hr after electroporation of control construct (white column) or conditional Sox9 expression construct (red column) and 24 hr after EdU administration at E14.5. (C–F, J–L) Scale bars, 50 $\mu$m (C–F, K, L), 20 $\mu$m (J). (G–I, M) Two-tailed, unpaired *t*-test: *p<0.05, **p<0.01, ***p<0.001. Data are the mean of six embryos electroporated with control construct and six embryos electroporated with conditional Sox9 overexpression construct (G), four embryos electroporated with control construct and four embryos electroporated with conditional Sox9 overexpression construct (H) and three embryos electroporated with control construct and three embryos electroporated with conditional Sox9 overexpression construct (I, M), each from a different litter; for each embryo, two microscopic fields, each of 200–$\mu$m apical width, were counted, and the values obtained were averaged. Error bars represent SEM.

The online version of this article includes the following figure supplement(s) for figure 4:

**Figure supplement 1.** *In vitro* validation of the functionality of the conditional Sox9 expression construct using HEK293T cells.

**Figure supplement 2.** Validation of the Sox9 expression, elicited specifically in BP-genic aRGs and BPs upon *in utero* electroporation of the conditional Sox9 expression construct into the neocortex of Tis21-CreER^T2 mouse embryos.

**Figure supplement 3.** Analysis of Pax6-negative mouse BPs upon conditional Sox9 expression.

---

E15.5 (**Figure 4—figure supplement 2E**). For both conditions, all RFP-positive cells were confined to the germinal zones at E14.5 (**Figure 4—figure supplement 2D**), whereas at E15.5 some RFP-positive cells were observed in the CP (**Figure 4—figure supplement 2E**), consistent with these cells being newly generated neurons.

These findings provided a basis to investigate the effects of conditional Sox9 expression in mouse BPs on their proliferative capacity. We first examined the expression of Ki67, a marker of cycling cells, upon electroporation of the mouse neocortex at E13.5 (**Figure 4B**). Indeed, conditional Sox9 expression doubled the proportion of cells in the SVZ and intermediate zone (IZ) derived from the targeted cells (as revealed by RFP expression) that were cycling and hence were BPs, both upon analysis at E15.5 (**Figure 4C,C', D,D', G**) and at E14.5 (**Figure 4H**).

To corroborate the effect of conditional Sox9 expression on mouse BP proliferation, we analyzed the abundance of basal mitoses by performing immunofluorescence for phospho-histone H3 one day after electroporation at E13.5 (**Figure 4E,F,F'**). Conditional Sox9 expression markedly increased the proportion of the targeted cell-derived (RFP+) BPs that underwent mitosis (**Figure 4I**).

In mouse, the overwhelming majority of BPs in the embryonic lateral neocortex at mid-neurogenesis are bIPs, with bRG constituting only a minor fraction (**Arai et al., 2011**; **Florio and Huttner, 2014**; **Kalebic et al., 2019**; **Shitamukai et al., 2011**; **Vaid et al., 2018**; **Wang et al., 2011**). Immunofluorescence for phospho-vimentin at E15.5 did not provide evidence that a noteworthy fraction of the targeted cell-derived (RFP+) mitotic BPs observed upon conditional Sox9 expression exhibited a prominent radial process (**Figure 4J**). This suggested that the Sox9-induced increase in the proliferation of mouse BPs pertained primarily to bIPs.

We sought to obtain additional evidence to support this notion. Mouse bRG have been shown to sustain Pax6 expression (**Shitamukai et al., 2011**; **Wang et al., 2011**), whereas mouse bIPs

downregulate this expression, although not necessarily to a zero level (*Arai et al., 2011*; *Englund et al., 2005*; *Fish et al., 2008*; *Hutton and Pevny, 2011*). Following *in utero* electroporation of control or conditional Sox9 expression construct at E13.5, we therefore analyzed targeted cell-derived (RFP+) mitotic (pVim+) BPs at E15.5 for Pax6 immunoreactivity (*Figure 4—figure supplements 3A,A'*). As at this developmental stage the majority of mouse bIPs are weakly Pax6-positive (*Hutton and Pevny, 2011*) and mouse bRG are also known to be Pax6-positive (*Shitamukai et al., 2011*; *Wang et al., 2011*), we confined our quantification to those BPs that were truly Pax6-negative, in order to be sure to quantify only bIPs. This revealed that conditional Sox9 expression doubled the proportion of the targeted cell-derived (RFP+) mitotic (pVim+) BPs that were Pax6-negative, that is bIPs (*Figure 4—figure supplement 3B*).

We next investigated whether the increase in mouse BPs upon conditional Sox9 expression was accompanied by an increase in cell cycle re-entry. To this end, we analyzed Ki67 immunofluorescence at E15.5 of mouse neocortex electroporated at E13.5 and subjected to EdU pulse-labeling at E14.5 (*Figure 4K,K', L,L'*). This revealed that conditional Sox9 expression significantly increased the proportion of targeted cell-derived (RFP+), EdU-containing cells in the SVZ plus IZ that were Ki67+ and hence were cycling BPs (*Figure 4M*). These data are consistent with the progeny of BPs exhibiting an increased ability to re-enter the cell cycle, that is to remain being BPs (as opposed to becoming postmitotic neurons).

## Conditional Sox9 expression in mouse BPs reduces Tbr2 expression and induces premature gliogenesis

BPs in the E13.5–15.5 mouse neocortex, that is bIPs, are typically neurogenic, undergoing symmetric consumptive division that generates two neurons (*Haubensak et al., 2004*; *Miyata et al., 2004*; *Noctor et al., 2004*), and express the transcription factor Tbr2 (*Figure 1G,J*; *Englund et al., 2005*; *Kowalczyk et al., 2009*; *Pontious et al., 2008*; *Sessa et al., 2008*). We asked whether conditional Sox9 expression, in addition to increasing the proliferative capacity of mouse BPs, would alter their identity and fate. Tbr2 immunofluorescence of mouse neocortex electroporated at E13.5 revealed that conditional Sox9 expression decreased the abundance of Tbr2-positive cells at E15.5 (*Figure 5A,B*), especially in the SVZ (*Figure 5A', B'*). Specifically, conditional Sox9 expression reduced the proportion of targeted cell-derived (RFP+) BPs in the SVZ that expressed Tbr2 to half (*Figure 5C*). These findings provided a first indication that conditional Sox9 expression in mouse BPs may alter their identity and fate, possibly reducing their commitment to neurogenesis.

In light of the involvement of Sox9 in the switch of NPCs from neurogenesis to gliogenesis reported previously for the developing mouse cerebellum and spinal cord (*Finzsch et al., 2008*; *Kang et al., 2012*; *Molofsky et al., 2013*; *Stolt et al., 2003*; *Stolt and Wegner, 2010*; *Vong et al., 2015*; *Wegner and Stolt, 2005*), we explored a potential role of Sox9 in gliogenesis in the developing neocortex. We first examined the Sox9-positive BPs in the ISVZ and OSVZ of the developing ferret neocortex (see *Figure 1B,E,H,K*) for the expression of Olig2, a transcription factor implicated in gliogenesis (*Marshall et al., 2005*; *Rowitch and Kriegstein, 2010*; *Zhou et al., 2001*). Double immunofluorescence for Sox9 and Olig2 of E40 ferret neocortex revealed scattered double-positive cells in the ferret ISVZ and OSVZ (see *Figure 5—figure supplement 1A and A'*), which upon quantification amounted to less than one fifth of the Sox9-positive cells in these germinal zones at E40/P1 (*Figure 5—figure supplement 1B*). Given that essentially all Sox9-positive cells in the E40/P1 ISVZ and OSVZ of ferret neocortex are Sox2-positive (*Figure 1E*, *Figure 5—figure supplement 1A,A', C*) and hence BPs, we conclude that only a minority of these progenitors are committed to gliogenesis at the developmental stages studied.

We next examined whether mouse BPs conditionally expressing Sox9 include gliogenic precursor cells. No Olig2 immunoreactivity was detected in the E15.5 neocortical wall by immunofluorescence upon control electroporation at E13.5 (*Figure 5D*), in line with the onset of neocortical gliogenesis in mouse occurring later, that is around E18.5 (*Miller and Gauthier, 2007*). Remarkably, however, upon conditional Sox9 expression for two days, many Olig2-positive cells were observed in the SVZ and IZ (*Figure 5E,E'*). Quantification of Olig2 and RFP double-positive cells revealed that 22% of the targeted cell-derived (RFP+) BPs in the mouse E15.5 SVZ and IZ had adopted a gliogenic identity (*Figure 5F*).

We examined whether Olig2 expression was related to the level of Sox9 expression. To this end, we performed *in utero* electroporation of the neocortex of tamoxifen-treated E13.5 *Tis21*-CreER^T2^

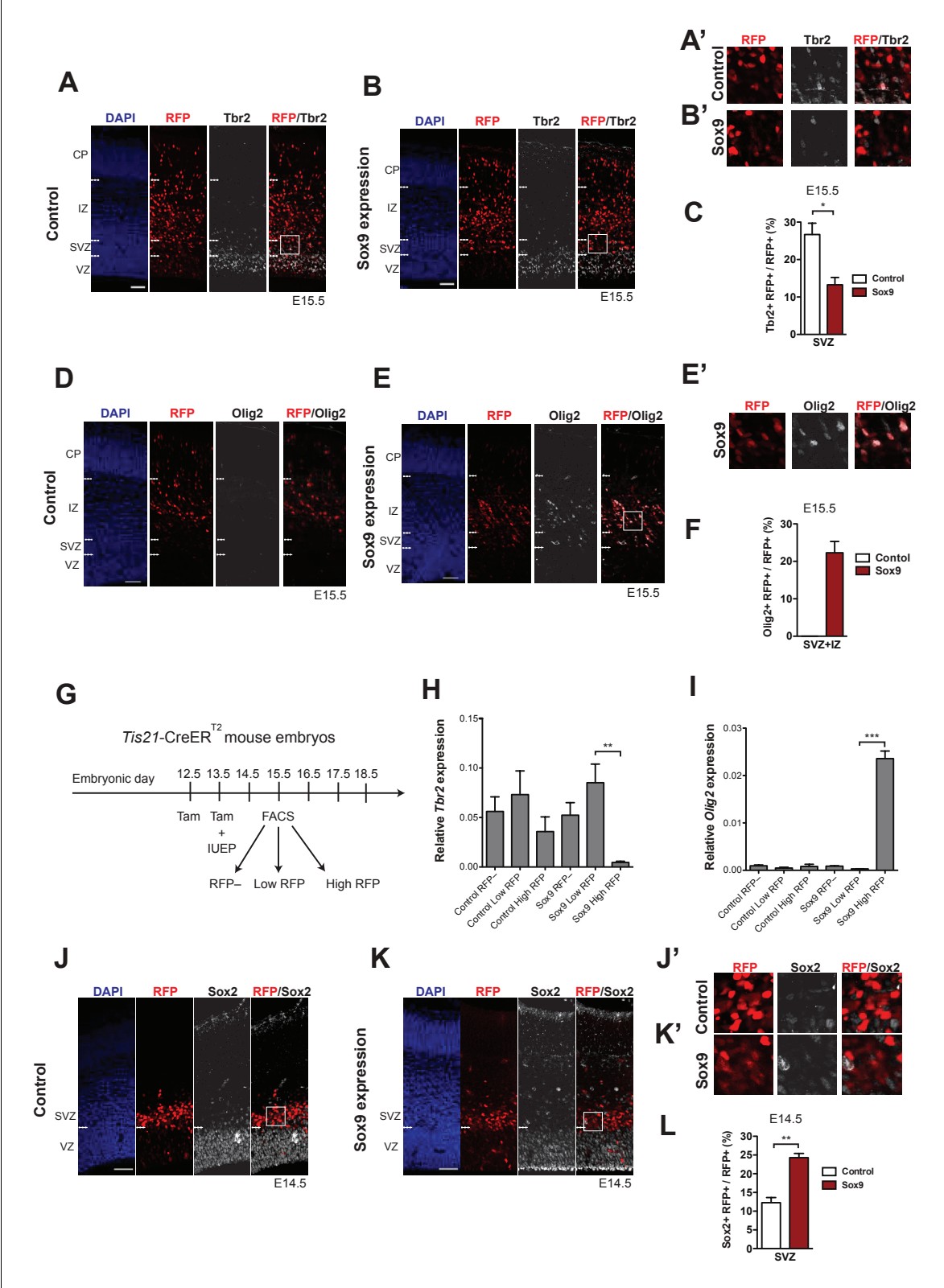

**Figure 5.** Conditional Sox9 expression in mouse BPs represses Tbr2 expression and induces premature gliogenesis in a dose-dependent manner. Heterozygous *Tis21*-CreER[T2] mouse embryos received tamoxifen administration at E12.5 and E13.5 and were subjected to *in utero* electroporation of the neocortex at E13.5 with either control construct or conditional Sox9 expression construct, followed by immunostaining analyses either 24 hr (J–L) or 48 hr (A–F) later. (A, B) Double immunofluorescence for RFP (red) and Tbr2 (white), combined with DAPI staining (blue), 48 hr after electroporation of

*Figure 5 continued on next page*

Figure 5 continued

control construct (A) or conditional Sox9 expression construct (B) (see *Figure 4A*). Dashed lines indicate the borders between VZ, SVZ, IP and CP. Boxed areas of the SVZ in (A) and (B) are shown at higher magnification in (A′) and (B′) respectively. (C) Quantification of the percentage of RFP-positive nuclei in the SVZ that are Tbr2-positive, 48 hr after electroporation of control construct (white column) or conditional Sox9 expression construct (red column). (D, E) Double immunofluorescence for RFP (red) and Olig2 (white), combined with DAPI staining (blue), 48 hr after electroporation of control construct (D) or conditional Sox9 expression construct (E). Dashed lines indicate the borders between VZ, SVZ, IP and CP. Boxed area of IZ in (E) is shown in higher magnification in (E′). (F) Quantification of the percentage of RFP-positive nuclei in the SVZ plus IZ that are Olig2-positive, 48 hr after electroporation of control construct (white column) or conditional Sox9 expression construct (red column). (G) Workflow of tamoxifen administration (Tam) at E12.5 and E13.5, *in utero* electroporation (IUEP) of the neocortex at E13.5, and FACS at E15.5 followed by qPCR analyses of RFP-positive cells yielding the data shown in (H) and (I), using heterozygous *Tis21*-CreER$^{T2}$ mouse embryos. (H, I) Quantification of *Tbr2* (H) and *Olig2* (I) mRNA levels relative to the *Gapdh* mRNA level by qPCR analysis, in RFP-negative, low-level RFP-expressing and high-level RFP-expressing cell populations isolated by FACS (see G) 48 hr after electroporation of control construct (left three columns) or conditional Sox9 expression construct (right three columns). (J, K) Double immunofluorescence for RFP (red) and Sox2 (white), combined with DAPI staining (blue), 24 hr after electroporation of control construct (J) or conditional Sox9 expression construct (K). Dashed lines indicate border between VZ and SVZ. Boxed areas in SVZ in (J) and (K) are shown at higher magnification in (J′) and (K′), respectively. (L) Quantification of the percentage of RFP-positive nuclei in the SVZ that are Sox2-positive, 24 hr after electroporation of control construct (white column) or conditional Sox9 expression construct (red column). (A, B, D, E, J, K) Scale bars, 50 µm. (C, H, I, L) Two-tailed, unpaired *t*-test: *p<0.05, **p<0.01, ***p<0.001. Data are the mean of 3 (C, F, L), 5 (H) and 4 (I) embryos, each from a different litter; for each embryo in (C, F, L), two microscopic fields, each of 200 µm apical width, were counted, and the values obtained were averaged. Error bars represent SEM.

The online version of this article includes the following figure supplement(s) for figure 5:

**Figure supplement 1.** Gliogenic potential of Sox9-expressing cNPCs in the germinal zones of developing ferret neocortex.

**Figure supplement 2.** Isolation of RFP-negative, low-level RFP-expressing and high-level RFP-expressing cNPC subpopulations by FACS and determination of *Sox9* mRNA levels in the six subpopulations.

mouse embryos with either the control RFP-expressing construct or the conditional Sox9 expression construct, dissociated the neocortical cells at E15.5, and isolated RFP+ cells by FACS (*Figure 5G*). With this approach, we obtained three cell populations each from control- and Sox9-electroporated neocortex, referred to as RFP–, Low RFP and High RFP (*Figure 5—figure supplement 2A,B*). We first examined *Sox9* mRNA levels by quantitative PCR on these sorted cell populations and observed that relative *Sox9* expression was highest in the High RFP population from Sox9-electroporated neocortex and was much lower in the Low RFP population from the same sample (*Figure 5—figure supplement 2C*). The other sorted cell populations (RFP–, Low RFP and High RFP of control, RFP– from Sox9-electroporated neocortex) showed almost no *Sox9* expression (*Figure 5—figure supplement 2C*). We next analyzed mRNA levels for Tbr2 and Olig2 and found that relative expression of Tbr2 mRNA was almost completely repressed (*Figure 5H*) and relative expression of Olig2 mRNA was specifically induced (*Figure 5I*) in the High RFP population of Sox9-electroporated neocortex. In contrast, the other sorted cell populations showed Tbr2 mRNA expression but no detectable Olig2 mRNA expression (*Figure 5H and I*). We conclude that high Sox9 levels in mouse BPs induce a switch in their cell fate to gliogenesis by inducing Olig2 mRNA and repressing Tbr2 mRNA expression.

The Sox9-induced, dose-dependent switch of BPs to a gliogenic fate raised the question whether all BPs had lost their neuronal progenitor identity. To answer this question, we immunostained control- and Sox9-electroporated neocortex for Sox2, at E14.5 (*Figure 5J and K*). Specifically, we examined the expression of Sox2 in the SVZ (*Figure 5J′ and K′*). Quantification of the percentage of the RFP+ cells in the SVZ that were Sox2-positive revealed an increase upon Sox9 expression (*Figure 5L*). Given that upon conditional Sox9 expression at E13.5, 10–15% of the RFP+ cells in the SVZ are still positive for Tbr2 at E15.5 (*Figure 5C*), this observation suggests that the increased number of BPs consist of a mix of neurogenic and gliogenic progenitors.

## Conditional Sox9 expression in mouse BPs mainly upregulates transcription of ECM components

In light of the findings described so far, it was important to obtain a global view of the transcriptional effects of conditional Sox9 expression in mouse BPs. To this end, we compared their transcriptomes upon electroporation of tamoxifen-treated E13.5 *Tis21*-CreER$^{T2}$ mouse neocortex with either the control or the conditional Sox9 expression construct, followed by dissociation of the electroporated tissue at E15.5, isolation of the High RFP cells by FACS (*Figure 5—figure supplement 2*), and

RNA sequencing (*Figure 6A*). We first assessed the difference between control- and Sox9-electroporated samples by principal component analysis (PCA), which revealed the two groups of samples to be distinct (*Figure 6—figure supplement 1A*). We then assembled a sample distance matrix of all expressed genes and observed that the conditionally Sox9 expressing samples clustered separately from the control samples (*Figure 6—figure supplement 1B*).

In order to identify the genes that were affected by conditional Sox9 expression, we performed differential gene expression (DGE) analysis by comparing the expression levels of only protein-encoding genes in conditionally Sox9 expressing vs control High RFP cells. This yielded 749 downregulated and 870 upregulated protein-encoding genes upon Sox9 expression (q < 0.01) (*Figure 6B*, *Supplementary file 1*). To gain insight into the cell biological processes affected by conditional Sox9 expression, we examined the 870 genes upregulated in Sox9-expressing cells by gene ontology (GO) term enrichment analysis for biological process (*Figure 6C*) and cellular component (*Figure 6D*). This analysis revealed that the highest enrichment scores most frequently included genes related to the ECM (*Figure 6C and D*). Similarly, KEGG pathway analysis using the same gene set as input yielded the highest enrichment scores for ECM production, organization and degradation (*Figure 6E*). GO term enrichment and KEGG pathway analyses indicated that 203 of the 870 genes upregulated in Sox9-expressing cells were related to the ECM (*Figure 6C–E*, *Supplementary file 3*).

In addition to the upregulation of expression of ECM components, conditional Sox9 expression induced the expression of gliogenic genes such as *Olig2* and *S100b* (*Figure 6—figure supplement*

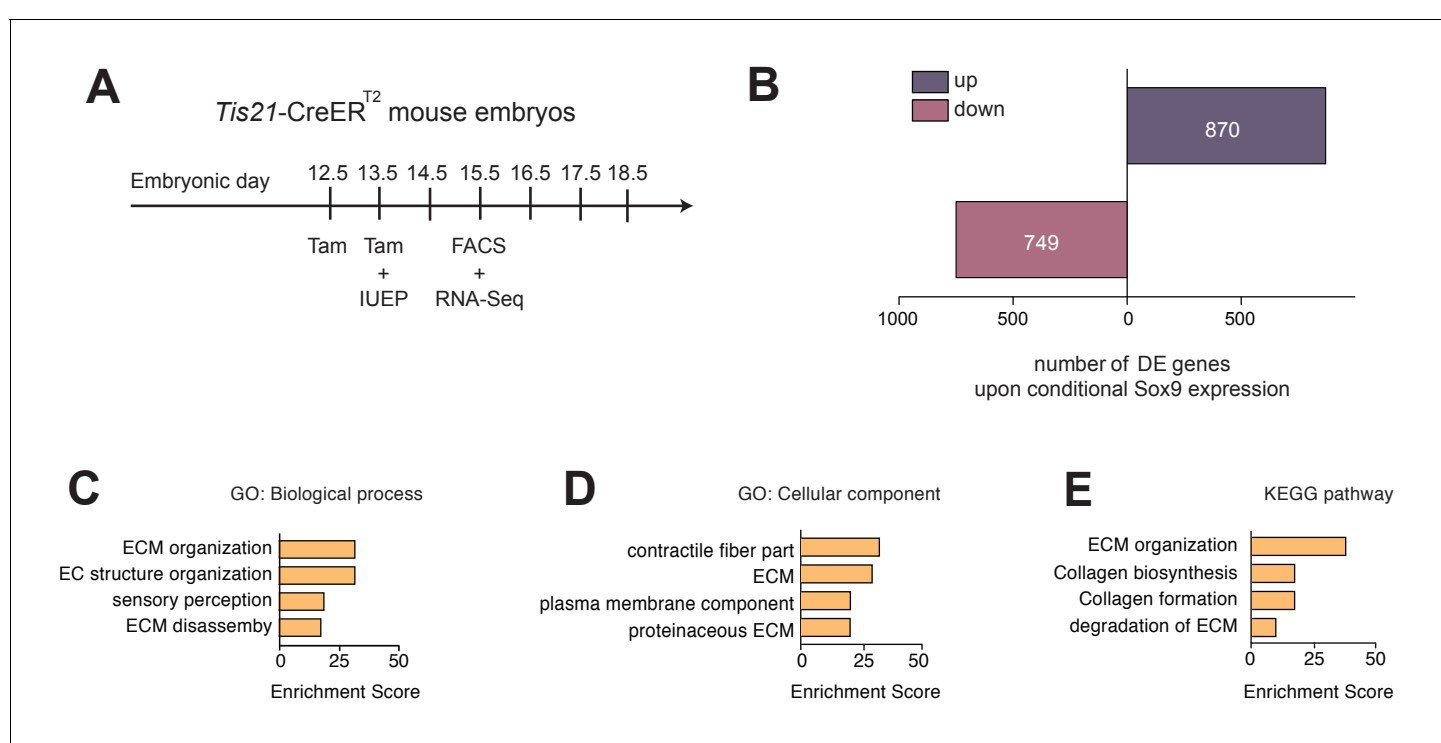

**Figure 6.** mRNAs upregulated upon conditional Sox9 expression in mouse BPs are mainly related to ECM. (A) Workflow of tamoxifen administration (Tam) at E12.5 and E13.5, *in utero* electroporation (IUEP) of the neocortex at E13.5 with either control construct or conditional Sox9 expression construct, and FACS at E15.5 followed by RNA-Seq analysis of high-level RFP-expressing cells yielding the data shown in (B–E), using heterozygous *Tis21*-CreER[T2] mouse embryos. (B) Number of mRNAs of protein-encoding genes differentially expressed (DE) 48 hr after electroporation of conditional Sox9 expression construct as compared to control construct (see *Figure 3A*); up, upregulated; down, downregulated. (C–E) Gene ontology (GO) term enrichment analyses for biological process (C) and cellular component (D), and KEGG pathway analysis (E), using as input the set of 870 genes upregulated upon conditional Sox9 expression. The top four enriched terms/pathways (p<0.01) are shown.

The online version of this article includes the following figure supplement(s) for figure 6:

**Figure supplement 1.** Principal component analysis and hierarchical clustering of electroporated cNPCs.

**Figure supplement 2.** mRNAs of gliogenesis markers are upregulated upon conditional Sox9 expression.

2A and B), corroborating the effect of conditional Sox9 expression on gliogenesis observed by immunohistochemistry (Figure 5E and F).

## Conditional Sox9 expression induces cell non-autonomous BP proliferation resulting in increased neuron production, especially of upper-layer neurons

ECM components have been reported to increase the proliferative capacity of BPs, and cell non-autonomous effects have been considered in this context (Fietz et al., 2010; Fietz et al., 2012; Florio et al., 2015; Long et al., 2016; Long and Huttner, 2019; Stenzel et al., 2014). In light of this, we examined if upregulated ECM production had an effect on the progeny in the SVZ derived from non-electroporated cells. To this end, we analyzed the immunofluorescence of electroporated neocortex to compare control and conditional Sox9 expression with regard to (i) Ki67-positive cells in the SVZ that were RFP-negative, that is progeny derived from non-electroporated cells (Figure 7A), (ii) RFP-negative abventricular mitoses (Figure 7B), and (iii) RFP-negative Sox2-positive nuclei in the SVZ (Figure 7C). Quantification per unit area of the numbers of Ki67-positive and RFP-negative nuclei in SVZ+IZ (Figure 7D), of PH3-positive and RFP-negative abventricular mitoses (Figure 7E), and of Sox2-positive and RFP-negative nuclei in SVZ+IZ (Figure 7F) revealed significant increases for all three cell populations in conditionally Sox9-expressing neocortex. These findings indicate that conditional Sox9 expression in mouse BPs not only increases their proliferation cell-autonomously (see Figure 4), but also results in a cell non-autonomous stimulation of BP proliferation.

We next investigated the outcome of the increased cell non-autonomous BP proliferation on neuron production. Specifically, we performed immunofluorescence for NeuN at E17.5 of mouse neocortex electroporated at E13.5 (Figure 7G and H), and found that the percentage of NeuN-positive nuclei in the CP was significantly increased upon conditional Sox9 expression (Figure 7I). (The finding that only about half of the nuclei in the CP were NeuN-positive in the control presumably reflects the fact that NeuN only labels mature neurons, but not neurons whose maturation process is not yet complete [Mullen et al., 1992]). Further examination of the NeuN-positive nuclei in the CP showed that conditional Sox9 expression in BPs resulted in a significant increase in the percentage of nuclei that were RFP-negative and NeuN-positive (Figure 7K); there was also a slight increase in the percentage of RFP-positive nuclei that were NeuN-positive, which however was not statistically significant, possibly reflecting the already high (>70%) percentage value in the control condition (Figure 7J). Taken together, we conclude that conditional Sox9 expression in mouse BPs results in increased neuron production from BPs whose proliferation was stimulated in a cell non-autonomous manner.

We further dissected the effects of conditional Sox9 expression in mouse BPs with regard to the identity of the neurons produced (Lodato and Arlotta, 2015). To this end, we carried out immunofluorescence for the upper-layer marker Satb2 (Britanova et al., 2008; Leone et al., 2015) and the deep-layer marker Tbr1 (Hevner et al., 2001; Molyneaux et al., 2007) at E17.5 of mouse neocortex electroporated at E13.5 and subjected to EdU pulse-labeling at E14.5 (Figure 7L and M, Figure 7—figure supplement 1A and B). On the one hand, conditional Sox9 expression in BPs resulted in a significant, more than two-fold increase in the number EdU-containing (EdU+) cells in the CP that were Satb2-positive and RFP-negative, that is progeny derived from of non-electroporated cells (Figure 7O). On the other hand, only a slight and not statistically significant increase in the proportion of EdU-containing (EdU+) RFP-positive (RFP+) cells that were Satb2-positive was observed upon conditional Sox9 expression in BPs (Figure 7N). In contrast, the proportion of EdU-containing (EdU+) RFP-positive (RFP+) cells that were Tbr1-positive (Tbr1+) was found to be massively decreased upon conditional Sox9 expression in BPs (Figure 7—figure supplement 1C). We conclude that upon conditional Sox9 expression in mouse BPs the production of upper-layer neurons is increased and that this increase involves the non-targeted BPs whose proliferation is stimulated in a cell non-autonomous manner.

We complemented these data by investigating the distribution of RFP-positive cells across the CP at E18.5, with the CP divided into 10 bins of equal size, comparing control and conditionally Sox9-expressing mouse neocortex electroporated at E13.5. (Figure 8A). Quantification of the percentage of RFP+ cells in each bin showed that upon conditional Sox9 expression in BPs, the relative distribution of the RFP+ cells was shifted towards the upper layers of the CP, with a significant decrease in

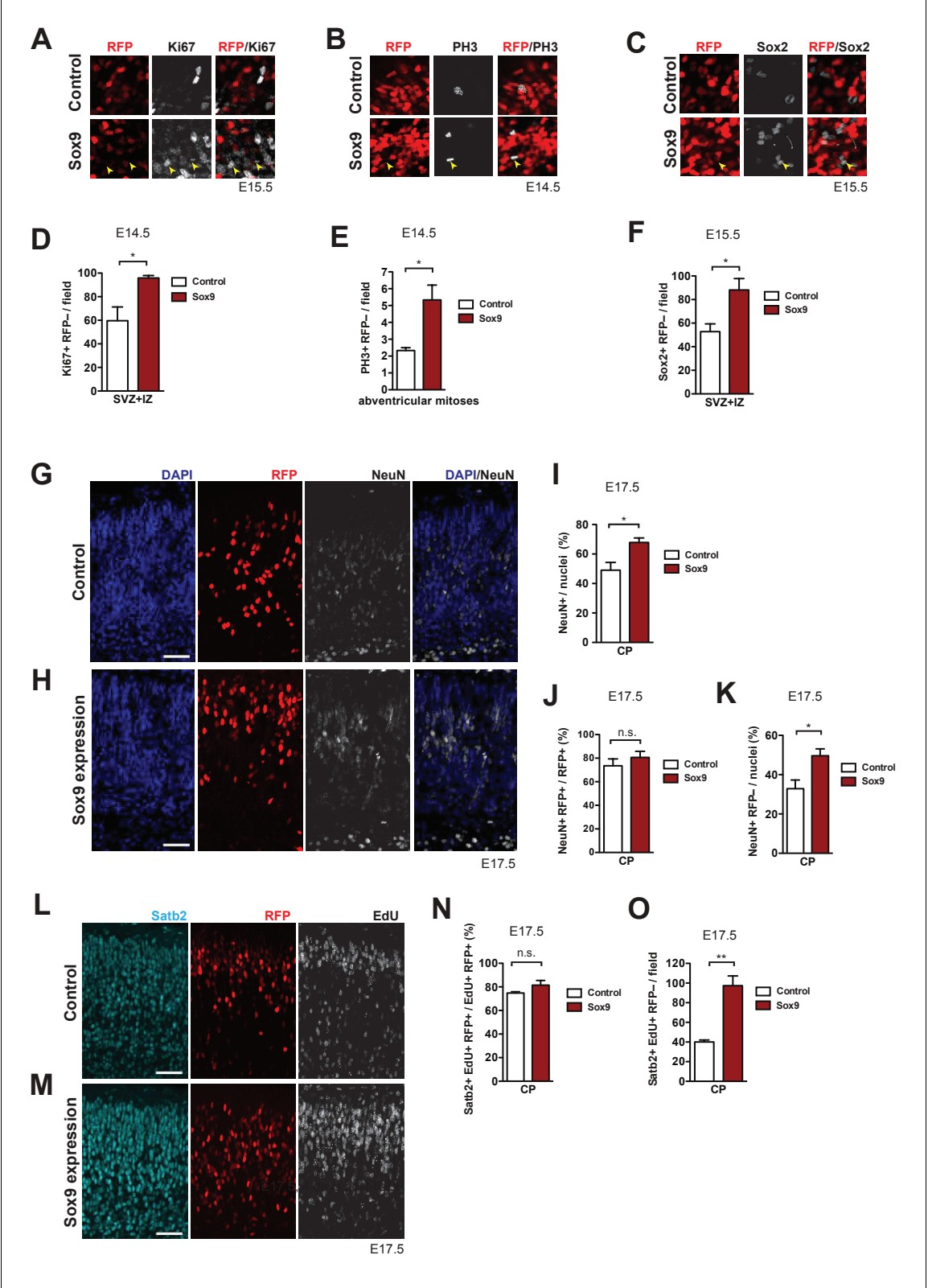

**Figure 7.** Conditional Sox9 expression in mouse BPs induces cell non-autonomous proliferation and neuron production. Heterozygous *Tis21*-CreER[T2] mouse embryos received tamoxifen at E12.5 and E13.5, were subjected to *in utero* electroporation of the neocortex with either control construct or conditional Sox9 expression construct (see *Figure 4A*) at E13.5, and subjected to immunostaining analyses of the neocortex at the time points indicated in (A–O). (A–C) High-magnification images of the SVZ showing double immunofluorescence for Ki67 (white) and RFP (red) (see yellow boxed

*Figure 7 continued on next page*

*Figure 7 continued*

areas in *Figure 4C, (D)* (A), phosphohistone H3 (PH3, white) and RFP (red) (see yellow boxed area in *Figure 4E, (F)* (B), and Sox2 (white) and RFP (red), 48 hr after electroporation of control or conditional Sox9 expression construct (C). Yellow arrowheads indicate BPs that are negative for RFP, but positive for Ki67 (A), PH3 (B) or Sox2 (C). (D) Quantification of the number of Ki67-positive, RFP-negative nuclei in the SVZ plus IZ per microscopic field of 200 μm apical width, 24 hr after electroporation of control construct (white column) or conditional Sox9 expression construct (red column). (E) Quantification of the number of abventricular, PH3-positive, RFP-negative mitoses per microscopic field of 200 μm apical width, 24 hr after electroporation of control construct (white column) or conditional Sox9 expression construct (red column). (F) Quantification of the number of Sox2-positive, RFP-negative nuclei in the SVZ plus IZ per microscopic field of 200 μm apical width, 48 hr after electroporation of control construct (white column) or conditional Sox9 expression construct (red column). (G, H) Double immunofluorescence for RFP (red) and NeuN (white), combined with DAPI staining (blue), in the CP 4 days after electroporation of control construct (G) or conditional Sox9 expression construct (H). (I) Quantification of the percentage of nuclei (identified by DAPI staining) in the CP that are NeuN-positive, 4 days after electroporation of control construct (white column) or conditional Sox9 expression construct (red column). (J) Quantification of the percentage of RFP-positive nuclei in the CP that are NeuN-positive, 4 days after electroporation of control construct (white column) or conditional Sox9 expression construct (red column). (K) Quantification of the percentage of nuclei (identified by DAPI staining) in the CP that are NeuN-positive and RFP-negative, 4 days after electroporation of control construct (white column) or conditional Sox9 expression construct (red column). (L, M) Triple (immuno)staining for Satb2 (cyan), RFP (red) and EdU (white) in the CP, 4 days after electroporation of control construct (L) or conditional Sox9 expression construct (M). A single pulse of EdU was administered at E14.5, that is 24 hr after electroporation and 3 days prior to analysis. (N) Quantification of the percentage of EdU and RFP double-positive nuclei in the CP that are Satb2-positive, that is the percentage of Satb2-positive neurons in the progeny of the targeted, EdU-labeled cNPCs, 4 days after electroporation of control construct (white column) or conditional Sox9 expression construct (red column) and 3 days after EdU administration at E14.5. (O) Quantification of the number of Satb2 and EdU double-positive, RFP-negative nuclei in the CP per microscopic field of 200 μm width, 4 days after electroporation of control construct (white column) or conditional Sox9 expression construct (red column) and 3 days after EdU administration at E14.5. (G, H, L, M) Scale bars, 50 μm. (D–F, I–K, N, O) Two-tailed, unpaired *t*-test: *p<0.05, **p<0.01, n.s. not significant. Data are the mean of 3 (E, F, I–K, N, O) and 4 (D) embryos, each from a different litter; for each embryo, two microscopic fields, each of either 200 μm (D–F, N, O) or 150 μm (I–K) width were counted, and the values obtained were averaged. Error bars represent SEM.

The online version of this article includes the following figure supplement(s) for figure 7:

**Figure supplement 1.** Conditional Sox9 expression results in a decrease of Tbr1-positive neurons in the progeny of the targeted, EdU-labeled mouse BPs.

---

the percentage of RFP+ cells in the lower layers of the CP (specifically bin number 2, *Figure 8B*) and an increase in the percentage of RFP+ cells in the upper layers of the CP (specifically bin number 7, *Figure 8B*). Consistent with these data and with the observed decrease in the proportion of EdU-containing RFP-positive cells that were Tbr1-positive at E17.5 upon conditional Sox9 expression (*Figure 7—figure supplement 1C*), the proportion of all RFP+ cells in the CP that were Tbr1+ at E18.5 was significantly decreased upon conditional Sox9 expression (*Figure 8C,C', D,D' and G*). Concomitant with this decrease, there was a small, albeit not statistically significant, increase in the proportion of RFP+ cells in the CP at E18.5 that were Satb2+ upon conditional Sox9 expression (*Figure 8E, E' F, F' and H*). Again, the lack of statistical significance likely reflected the already high (>70%) percentage value in the control condition. These data were consistent with the slight but not statistically significant increase in the proportion of EdU-containing RFP-positive cells in the CP at E17.5 that were Satb2-positive upon conditional Sox9 expression in BPs (*Figure 7N*). Further analysis of the Satb2-positive nuclei in the CP showed that conditional Sox9 expression in BPs resulted in a significant increase in the abundance (i) of nuclei that were RFP-negative and Satb2-positive (*Figure 8I*) and (ii) of the total Satb2-positive nuclei in the CP (*Figure 8J*). These findings corroborate the notion that the increase in Satb2-positive neurons upon conditional Sox9 expression in BPs pertains mostly to progeny derived from non-electroporated cells.

## ECM component laminin 211 promotes BP proliferation

Finally, in light of both, the increased expression of specific ECM components and the increase in BP proliferation, upon conditional Sox9 expression, we sought to directly demonstrate that selected candidate ECM components can promote mouse BP proliferation. To this end, we incubated organotypic slices of E14.5 wildtype mouse neocortex in the absence and presence of either recombinant collagen IV or recombinant laminin 211 (laminin α2, β1, γ1). Collagen IV and laminin 211 were chosen as representative members of the collagen and laminin families of proteins, respectively, which have previously been implicated in cortical progenitor self-renewal and proliferation (*Fietz et al., 2012*). Of note, collagen IV mRNAs have been found to be expressed not only in the fetal human VZ but also ISVZ and OSVZ (*Fietz et al., 2012*), in line with the proliferative capacity of the cortical

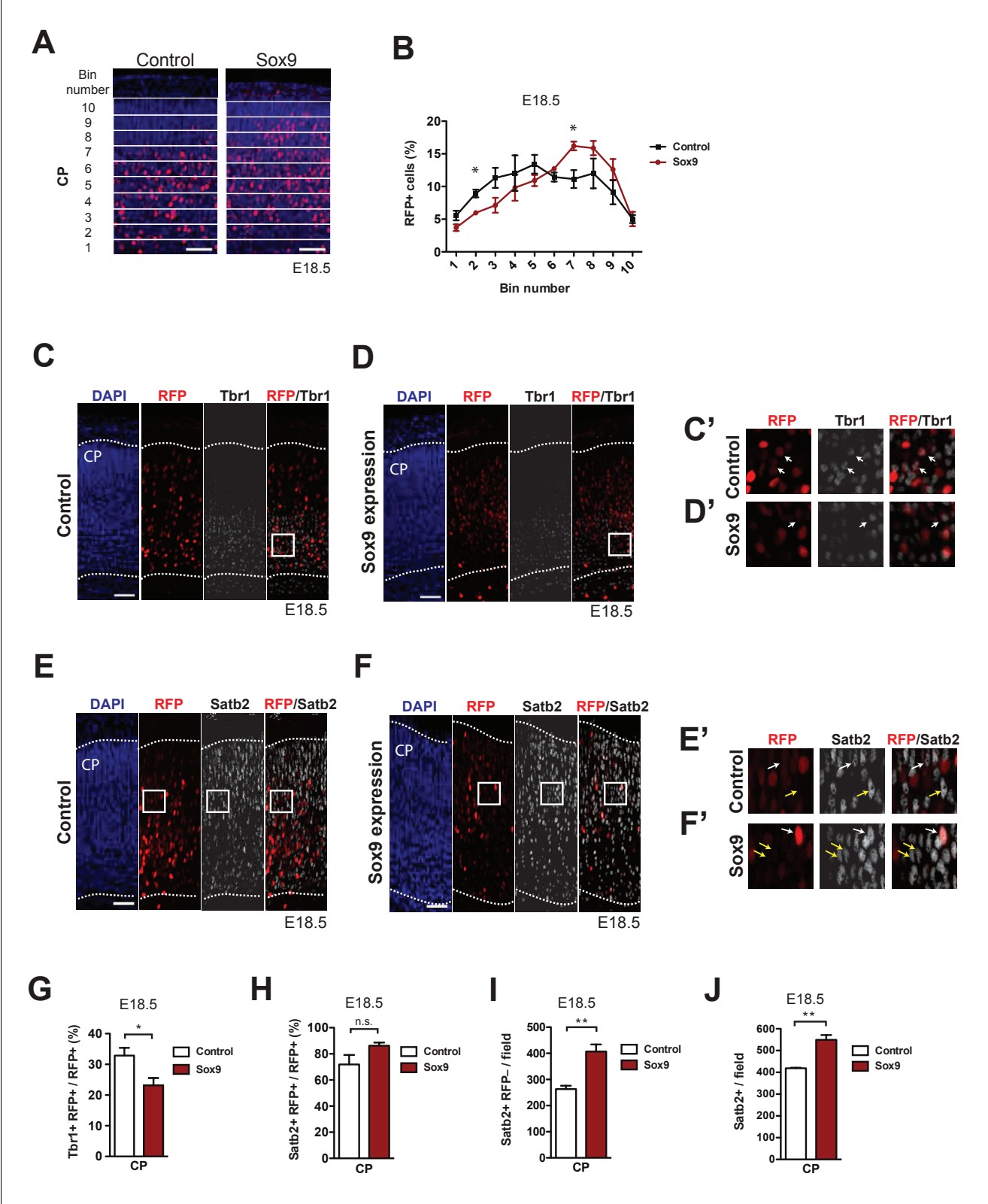

**Figure 8.** The neurons derived from mouse BPs targeted by conditional Sox9 expression are preferentially destined to the upper layers of the neocortex. Heterozygous *Tis21*-CreER[T2] mouse embryos received tamoxifen at E12.5 and E13.5, were subjected to *in utero* electroporation of the neocortex at E13.5 with either control construct or conditional Sox9 expression construct, and subjected to immunostaining analyses at E18.5. (**A**) Immunofluorescence for RFP (red), combined with DAPI staining (blue), in the CP 5 days after electroporation of control or conditional Sox9 expression

*Figure 8 continued on next page*

*Figure 8 continued*

construct (see *Figure 3A*). For the quantification of the distribution of RFP-positive nuclei (see B), images were divided into ten bins of equal size (numbered 1–10 from apical to basal) that span the entire CP. (B) Quantification of the percentage of the RFP-positive nuclei in the CP that are found in each bin (see A), 5 days after electroporation of control construct (black line) or conditional Sox9 expression construct (red line). (C, D) Double immunofluorescence for RFP (red) and Tbr1 (white), combined with DAPI staining (blue), in the CP 5 days after electroporation of control (C) or conditional Sox9 expression (D) construct. Dotted lines delineate the apical and basal borders of the CP. White boxed areas of the CP in (C) and (D) are shown at higher magnification in (C') and (D'), respectively. (E, F) Double immunofluorescence for RFP (red) and Satb2 (white), combined with DAPI staining (blue), in the CP 5 days after electroporation of control (E) or conditional Sox9 expression (F) construct. Dotted lines delineate the apical and basal borders of the CP. White boxed areas of the CP in (E) and (F) are shown at higher magnification in (E') and (F'), respectively. (G, H) Quantification of the percentage of the RFP-positive nuclei in the CP that are Tbr1-positive (G) and Satb2-positive (H), 5 days after electroporation of control construct (white columns) or conditional Sox9 expression construct (red columns). (I) Quantification of the number of Satb2-positive, RFP-negative nuclei in the CP per microscopic field of 200 µm width, 5 days after electroporation of control construct (white column) or conditional Sox9 expression construct (red column). (J) Quantification of the total number of Satb2-positive nuclei in the CP per microscopic field of 200 µm width, 5 days after electroporation of control construct (white column) or conditional Sox9 expression construct (red column). (A, C– F) Scale bars, 50 µm. (B) One-way ANOVA: p<0.0001; two-tailed, unpaired *t*-test performed separately for each bin pair: *p<0.05. Data are the mean of 3 embryos, each from a different litter; for each embryo, two microscopic fields, each of 200 µm width, were counted, and the values obtained were averaged. Error bars represent SEM. (G– J) Two-tailed, unpaired *t*-test: *p<0.05, **p<0.01, n.s. not significant. Data are the mean of 3 embryos, each from a different litter; for each embryo, two microscopic fields, each of 150 µm width, were counted, and the values obtained were averaged. Error bars represent SEM.

progenitors therein. Accordingly, collagen IV mRNAs are found to be expressed in both aRG and bRG (*Florio et al., 2015*). Likewise, with regard to laminin 211, laminin α2 (i) has been shown to be expressed not only in the embryonic mouse VZ but also SVZ (*Lathia et al., 2007*) and, notably, in the human OSVZ (*Fietz et al., 2012*), as well as in human aRG and bRG (*Florio et al., 2015*); and (ii) to be involved in the regulation of AP proliferation (*Loulier et al., 2009*).

The present mRNA analyses provided additional data that warranted an examination of a potential role of laminin 211 and collagen IV in promoting mouse BP proliferation upon conditional Sox9 expression. Specifically, although the laminin α2 mRNA – because of its low FPKM values – was not included in our analysis of the protein-encoding genes upregulated upon conditional Sox9 expression, it did show a ≈two-fold increase upon conditional Sox9 expression (*Figure 9A*). In addition, the mRNAs for the beta and gamma chains required to form a functional laminin 211 heterotrimer, Lamb1 and Lamc1, were found to exhibit expression levels in both the control condition and upon conditional Sox nine expression in the mouse BP lineage that were consistent with the occurrence of laminin 211 in the embryonic mouse neocortex under these conditions (*Figure 9B,C*). Taken together, these data suggested to us that it would be worthwhile to examine a potential role of laminin 211 in promoting mouse BP proliferation. In this context, we would like to emphasize that laminin α2 is of course not the only ECM target of conditional Sox9 expression, and that laminin 211 was primarily used here as a proof-of-principle experiment to support the concept: conditional Sox9 expression –>increased expression of ECM components –>BP proliferation. Similar considerations pertained to the collagen 4α1 mRNA, one of the 870 genes upregulated upon conditional Sox nine expression (*Supplementary file 1*), which showed a > 3 fold increase in this condition (*Figure 9D*).

Following the treatment of organotypic slices of E14.5 wildtype mouse neocortex without or with recombinant collagen IV or laminin 211, our analysis of cell proliferation by immunofluorescence for Ki67 and phosphohistone H3 (*Figure 9E,E'*) revealed that laminin 211, but not collagen IV, caused a significant increase in the proportion of cycling cells (Ki67+) specifically in the SVZ, but not VZ (*Figure 9F*). In addition, the number of basal mitoses (PH3+) per unit area was strikingly increased by laminin 211, but not by collagen IV, whereas the number of apical mitoses was largely unaffected by either ECM component (*Figure 9G*). These findings provide strong support for the concept that conditional Sox9 expression in mouse BPs increases the expression of specific ECM components, notably laminins, which potentially promote BP proliferation in an autocrine (RFP-positive BPs, cell autonomous) and paracrine (RFP-negative BPs, cell non-autonomous) manner.

## Discussion

The present study establishes Sox9 as an important regulator of BP proliferation and cell fate. Remarkably, the four major effects of Sox9, that is, on (i) transcription, (ii) BP proliferation (*Figure 9— figure supplement 1A*), (iii) neuron production (*Figure 9—figure supplement 1C*), and (iv) BP cell

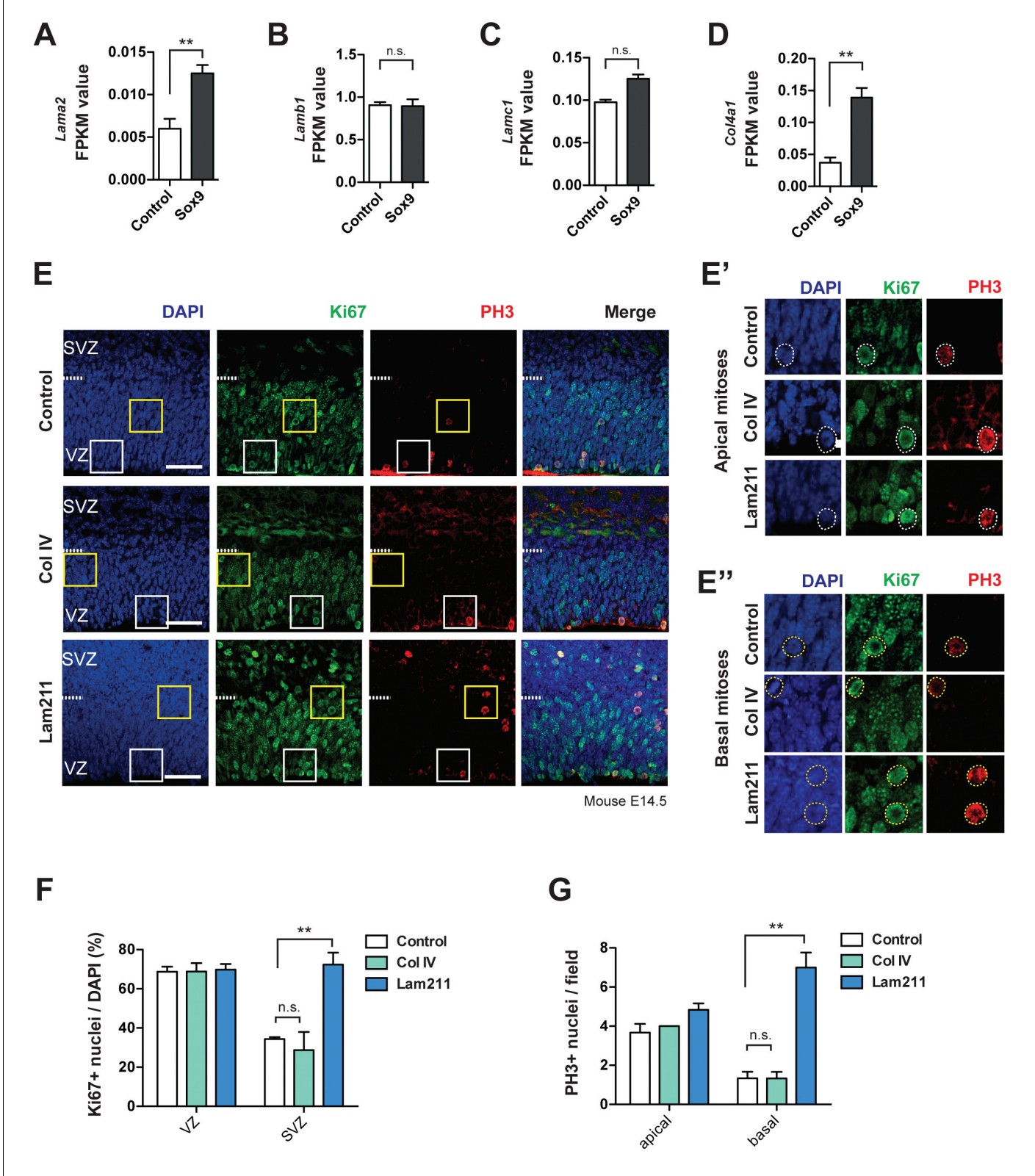

**Figure 9.** Laminin 211 induces BP proliferation in embryonic mouse neocortex. (**A–D**) FPKM values for *Lama2* mRNA (**A**), *Lamb1* mRNA (**B**), *Lamc1* mRNA (**C**) and *Col4a1* mRNA (**D**) in the high-level RFP-expressing cells at E15.5 upon electroporation of heterozygous *Tis21*-CreER^T2 mouse embryos with control and conditional Sox9 expression constructs at E13.5 (for details, see *Figure 6A* and legend). Two-tailed, unpaired *t*-test: **p<0.01, n.s. not significant. Data are the mean of transcriptomes of high-level RFP+ cells from 3 (control electroporated) and 4 (Sox9 electroporated) mouse embryonic

*Figure 9 continued on next page*

*Figure 9 continued*

neocortices, respectively. Error bars represent SEM. (E, E', E'') Double immunofluorescence for Ki67 (green) and phosphohistone H3 (PH3, red), combined with DAPI staining (blue), of mouse E14.5 organotypic slices of neocortex cultured for 24 hr without (control) or with collagen IV (Col IV) or laminin $\alpha_2\beta_1\gamma_1$ (Lam211). Ventricular surface is down, dashed lines indicate the border between the VZ and SVZ. Scale bars, 50 µm. White and yellow boxed areas are shown at higher magnification in (E') and (E'') and present examples of apical mitoses and basal mitoses, respectively; dashed circles indicate Ki67- and PH3-positive mitotic cNPCs. (F) Quantification of the percentage of the nuclei (identified by DAPI staining) that are Ki67-positive, in the VZ and SVZ of 24 hr control (white columns), Col IV-treated (green columns) and Lam211-treated (blue columns) E14.5 mouse neocortical organotypic slice cultures. (G) Quantification of the number of ventricular (apical) and abventricular (basal) PH3-positive mitoses, per microscopic field of 200 µm apical width, in 24 hr control (white columns), Col IV-treated (green columns) and Lam211-treated (blue columns) E14.5 mouse neocortical organotypic slice cultures. (F, G) Two-tailed, unpaired *t*-test, **p<0.01. Data are the mean of 3 embryos, each from a different litter; for each embryo, two microscopic fields, each of 200 µm apical width, were counted, and the values obtained were averaged. Error bars represent SEM. Note that the three data for apical mitoses upon collagen IV treatment were identical.

The online version of this article includes the following figure supplement(s) for figure 9:

**Figure supplement 1.** Cartoon illustrating the effects of conditional Sox9 expression in the mouse BP lineage on BP proliferation, fate choice and neuron output.

---

fate (*Figure 9—figure supplement 1B*), constitute a harmonic quartet of impacts, as is addressed one by one below.

## Sox9 is a major inducer of ECM components

Corroborating our considerations at the start of this study, that is that Sox9 – when expressed in the SVZ – could be a promising candidate to drive the expression of ECM components in BPs, our transcriptome analysis upon conditional Sox9 expression in mouse BPs revealed that, indeed, the 870 genes upregulated in Sox9-expressing BPs were predominantly related to the ECM by GO term enrichment analysis. The finding that Sox9 expression in BPs drives the expression of ECM components suggests an elegant explanation for the cell non-autonomous stimulation of BP proliferation by Sox9, as delineated below.

## Sox9 promotes BP proliferation cell-autonomously and non-autonomously

Comparison of embryonic mouse, embryonic ferret and fetal human neocortex showed that Sox9 expression in the SVZ correlated with the occurrence of BPs endowed with proliferative capacity, being detected in the ferret and human but not mouse SVZ. Further evidence in support of a positive role of Sox9 in the proliferation of ferret and human BPs was provided by the observations that these BPs were cycling and capable of cell cycle re-entry, and that they largely lacked expression of Tbr2, a marker that is often expressed in those BPs that are committed to neuron production rather than proliferation (*Arnold et al., 2008*; *Englund et al., 2005*; *Florio et al., 2015*; *Kowalczyk et al., 2009*; *Sessa et al., 2008*). Moreover, essentially all of the human bRG, the BP cell type particularly implicated in neocortical expansion (*Fietz et al., 2010*; *Florio and Huttner, 2014*; *Hansen et al., 2010*), were found to express Sox9. These findings extend previous data on the role of Sox9 in apical progenitor proliferation (*Scott et al., 2010*) and clearly suggest a possible involvement of Sox9 in BP proliferation.

Direct evidence for this notion came from two sets of experiments. First, the CRISPR/Cas9-mediated KO of Sox9 in embryonic ferret neocortex, which contains highly proliferative BPs, showed that Sox9 is actually required for BP proliferation in a gyrencephalic brain. Second, the conditional expression of Sox9 in mouse BPs, which normally have a low proliferative capacity, showed that Sox9 can promote BP proliferation also ectopically. The remarkable 70% reduction in Sox2+ cells in the ferret embryonic SVZ upon Sox9 KO further indicates a prominent role of Sox9 for the proliferative capacity of bRG, the key cell type involved in both the evolutionary expansion of the neocortex and the proper human neocortex development.

Strikingly, however, the BPs that showed increased proliferative capacity upon conditional Sox9 expression in embryonic mouse neocortex were not only the progeny of electroporated cells, but also the progeny of the non-electroporated cells. In other words, Sox9 promoted not only the proliferation of those mouse BPs in which it was expressed in a targeted manner, but also of neighboring BPs lacking Sox9 expression. Previous findings have implicated ECM components as stimulators of

cNPC proliferation that act via integrin signaling (*Ahmed et al., 2019*; *Arai et al., 2011*; *Fietz et al., 2010*; *Fietz et al., 2012*; *Florio et al., 2015*; *Hall et al., 2008*; *Long et al., 2016*; *Stenzel et al., 2014*). Moreover, the present observations demonstrate that Sox9 expression in BPs induces the expression of ECM components, some of which (like laminin 211) can promote BP proliferation. It therefore seems likely that the cell non-autonomous stimulation of mouse BP proliferation upon conditional Sox9 expression is due to ECM components that have been secreted from Sox9-expressing BPs and then act on neighboring BPs that themselves have not been targeted to express Sox9. If so, this interplay reflects how well the Sox9 effects on transcription and BP proliferation are orchestrated.

We do not know which genes whose expression is increased upon conditional Sox9 expression in mouse BPs are responsible for the cell-autonomous stimulation of BP proliferation by Sox9. Presumably, Sox9 acts in concert with other cell-intrinsic factors, including Sox2 and transcription factors with which it can heterodimerize such as Sox8 and Sox10 (*Huang et al., 2015*). In this context, it is interesting to note that Sox8 shows a similar expression pattern with regard to the neocortical germinal zones as Sox9, being expressed in embryonic mouse neocortex essentially only in the VZ but in fetal human neocortex in the VZ, ISVZ and OSVZ (*Fietz et al., 2012*). Hence, if a heterodimerization or concerted action of Sox9 with Sox8 (or with other factors exhibiting similar expression patterns) underlies the stimulation of BP proliferation, such differential expression between species could explain why the effects of Sox9 KO in embryonic ferret neocortex appear to be stronger than those of conditional Sox9 expression in embryonic mouse neocortex. In addition, given that the cell non-autonomous stimulation of BP proliferation upon Sox9 expression likely involves ECM components secreted from Sox9-expressing BPs, it seems possible that the mechanism underlying the cell-autonomous stimulation of BP proliferation by Sox9 also involves ECM components secreted from the Sox9-expressing BPs and acting on them in an autocrine fashion.

## Sox9 increases production of upper-layer neurons

The effects of conditional Sox9 expression in mouse BPs on neuron production added yet a third line of evidence for the orchestrated effects of Sox9. An increase in upper-layer neuron production is considered to be one of the hallmarks of neocortical expansion (*Hutsler et al., 2005*; *Molnár et al., 2006*). Accordingly, conditional Sox9 expression in mouse BPs resulted in this phenotype. Interestingly, however, whereas the Sox9-induced decrease in the production of deep-layer neurons was due to the effects of Sox9 in the Sox9-expressing BPs, the increase in the production of upper-layer neurons reflected an indirect effect via the BPs that themselves were not targeted to express Sox9. This illustrates how a synergistic effect (less deep-layer neurons, more upper-layer neurons) can be achieved via concerted direct and indirect effects of Sox9 on BPs.

## Sox9 switches BPs to gliogenesis

Finally, the quartet of harmonic impacts of Sox9 expression was completed by its effects on the fate of BPs. Specifically, not only did conditional Sox9 expression in mouse BPs reduce the proportion of Tbr2-positive BPs, consistent with the stimulation of BP proliferation by Sox9, but conditional Sox9 expression also induced Olig2 in a subset of BPs, indicative of switching them to gliogenesis. In the adult mouse brain, Sox9 has been reported to be expressed simultaneously by mature astrocytes and neurogenic SVZ progenitors (*Cheng et al., 2009*; *Martini et al., 2013*; *Nagao et al., 2014*; *Nait-Oumesmar et al., 2008*; *Sun et al., 2017*). In the embryonic mouse neocortex, Sox9-expressing aRG have been shown to be capable of generating neurons for all layers (*Kaplan et al., 2017*). Our findings are therefore consistent with the multi-faceted role of Sox9 in neurogenesis and gliogenesis (*Cheng et al., 2009*; *Kaplan et al., 2017*; *Nagao et al., 2014*; *Nagao et al., 2016*; *Selvaraj et al., 2017*; *Sun et al., 2017*).

Neocortical expansion is characterized not only by an increase in neurogenesis, but also in gliogenesis (*Rash et al., 2019*), which might be particularly important in the context of human brain evolution, as recent findings show that expression of the human-specific pro-proliferative gene *ARHGAP11B* leads to an increase in both upper-layer neurons and glia (*Kalebic et al., 2018*). In light of this, the effects of Sox9 expression highlight the concerted nature of its action on several key aspects of neocortex development and evolutionary expansion. Hence, when collectively considering its effects on transcription, BP proliferation and fate, and neuron production, Sox9 acting in the SVZ

shows many of the hallmarks expected for a transcription factor contributing to promote neocortical expansion.

## Materials and methods

### Mice

Embryos from C57Bl/6JOlaHsd mice at embryonic day (E) 13.5 and 14.5 were used as wild type. Conditional Sox9 expression experiments were carried out in heterozygous embryos of the *Tis21*-CreER$^{T2}$ transgenic mouse line, in which exon 1 of the *Tis21* gene was replaced with a CreER$^{T2}$ cassette (*Wong et al., 2015*). E0.5 was specified as the noon of the day when plugged females were observed. All mouse lines used in this study were congenic and kept in pathogen-free conditions at the Biomedical Services facility (BMS) of the Max Planck Institute of Molecular Cell Biology and Genetics, Dresden. All experiments utilizing mice were conducted in agreement with the German Animal Welfare Legislation after approval by the Landesdirektion Sachsen (licence TVV 05/2015).

### Ferrets

Timed-pregnant ferrets (*Mustela putorius furo*) were obtained from Euroferret (Copenhagen, Denmark) or Marshall BioResources (NY, USA) and housed at the BMS of MPI-CBG or BioCrea GmBH (Radebeul, Saxony, Germany). Observed mating date was set to E0. All experiments were performed in the dorsolateral telencephalon of ferret embryos, at a medial position along the rostro-caudal axis, in the prospective motor and somatosensory cortex, and were conducted in agreement with the German Animal Welfare Legislation after approval by the Landesdirektion Sachsen (licence TVV 21/2017).

### Human tissue

Human fetal neocortical tissue was provided by the Klinik und Poliklinik für Frauenheilkunde und Geburtshilfe, Universitätsklinikum Carl Gustav Carus, involving elective pregnancy terminations with informed written maternal consents and approval by the local University Hospital Ethical Committees. Human fetal neocortical tissue was also obtained from Novogenix Laboratories (Torrance, CA). The age of fetuses was assessed by ultrasound measurements of crown-rump length and other standard criteria of developmental stage determination.

### Plasmids

Plasmid pCAGGS-LoxP-membraneGAP43-GFP-LoxP-IRES-nRFP (referred as control construct) was generated as previously described (*Wong et al., 2015*) and kindly provided by Dr. Fong Kuan Wong. Plasmid pCAGGS-LoxP-membraneGAP43- GFP-LoxP-Sox9-IRES-nRFP (referred as conditional Sox9 expression construct) was generated by cloning the Sox9 coding sequence into the control construct 5' to the IRES sequence. The Sox9 coding sequence was amplified from FUW-TetO-Sox9 plasmid (Addgene, #41080) and cloned into the control plasmid using restriction enzyme XhoI (NEB, #R146L). pCAGGS-Cre plasmid was generated as described (*Kranz et al., 2010*) and kindly provided by Dr. JiFeng Fei.

Primers used to amplify the Sox9 coding sequence:

Sox9_F: 5'-TCTCGAGGCCGCCATGAATCTCCTGGACCCCTTC-3'
Sox9_R: 5'-TCTCGAGTCAGGGTCTGGTGAGCTGTGT-3'

### Cell culture

HEK293T cells were plated in 24-well plates with a density of $5 \times 10^4$ cells per well, and cultured in DMEM containing 10% fetal bovine serum. After 24 hr of culturing, the cells were transfected using Lipofectamine 2000 (Invitrogen) with 1 µg of conditional Sox9 expression plasmid, with 1 µg of control plasmid, or with either conditional Sox9 expression or control plasmid in combination with 1 µg of pCAGGS-Cre plasmid. After transfection, the cells were cultured for an additional 48 hr and fixed with 4% (wt/vol) paraformaldehyde in 120 mM phosphate buffer (pH 7.4) (referred to from here on simply as PFA) for 10 min at room temperature.

## CRISPR/Cas9 strategy and *in vitro* preparation of the Cas9/gRNA complex

The CRISPR/Cas9 approach used for obtaining a Sox9 KO in the embryonic ferret neocortex was similar to the approach established previously for the mouse embryonic neocortex (*Kalebic et al., 2016*). For control, a previously published gRNA targeting *LacZ* was used (*Kalebic et al., 2016*). For disruption of *Sox9*, the genomic sequence of ferret *Sox9* was analyzed for CRISPR/Cas9 target sites by Geneious 11 software (Biomatters), and two gRNAs were selected (sequences given in the *Figure 3—figure supplement 1A*). gRNAs were produced by *in vitro* transcription from a PCR product as previously described (*Kalebic et al., 2016*). Recombinant Cas9 protein (ToolGen) from *Streptococcus pyogenes* was used. Preparation of the recombinant Cas9 protein and gRNA complexes was performed as described previously (*Kalebic et al., 2016*).

## *In utero* electroporation of mouse embryos

*In utero* electroporations (IUEPs) were performed as previously described (*Shimogori and Ogawa, 2008*). Pregnant *Tis21*-CreER^T2 mice were treated via oral gavage with 2 mg tamoxifen dissolved in corn oil, at E12.5 and at E13.5. After tamoxifen administration, pregnant mice carrying E13.5 embryos were anesthetized with isofluorane. Then, 1.5 µg/µl of either control or conditional Sox9 expression plasmid DNA was mixed with 0.25% Fast Green FCF dye and was intraventricularly injected into each embryo with a glass microcapillary and electroporated by a series of 6 pulses (30V) with 1 ms intervals. Embryos were collected at 24 hr, 48 hr, 4 days or 5 days after electroporation.

## *In utero* electroporation of ferret embryos

*In utero* electroporation of ferret embryos as well as pre-operative and post-operative care of ferrets were performed as previously described (*Kalebic et al., 2018*). All the ferrets that underwent *in utero* electroporation underwent another surgery four days later that followed the same pre-operative care, anesthesia and analgesia as described previously (*Kalebic et al., 2018*). This second surgery was performed in order to remove the electroporated embryos by Caesarian section and to carry out a subsequent complete hysterectomy. Animals were kept at the BMS of the MPI-CBG for at least two weeks after the second surgery after which they were donated for adoption. Embryonic brains were dissected and PFA-fixed for the immunofluorescence analyses.

## EdU labeling

EdU was dissolved in PBS at a concentration of 1 mg/ml. For analyzing the cell cycle re-entry of Sox9-expressing progenitors, P0 ferret kits were injected intraperitoneally with 100 µl of EdU solution. P2 ferret kits were subjected to hypothermic anesthesia in crushed ice and sacrificed by intracardiac perfusion with 4% PFA at 37°C. In order to analyze the cell cycle re-entry of the progeny of electroporated progenitors in mouse embryos, 100 µl of EdU solution was intraperitoneally injected into pregnant, electroporated mice at E14.5. EdU fluorescence was detected using the Click-iT EdU Alexa Fluor 647 Imaging Kit (Invitrogen C10340) according to the manufacturer's instructions.

## Immunofluorescence

For immunofluorescence of transfected HEK293T cells, the cells were fixed on coverslips with 4% PFA at room temperature for 10 min and permeabilized with 0.3% Triton X-100 in PBS for 30 min, then quenched with 0.1 M glycine in 1x PBS for 30 min. Coverslips were then washed with 1x PBS containing 0.2% gelatin, an additional 300 mM NaCl, and 0.3% Triton X-100 and incubated with primary antibodies either for 3 hr at room temperature or overnight at 4°C, followed by incubation with secondary antibodies at room temperature for 1 hr. Cells were washed with PBS and mounted in Mowiol (Merck Biosciences).

In order to perform immunofluorescence on neocortex, the tissue was fixed with 4% PFA overnight at 4°C. The tissue was further processed for either vibratome or cyrosectioning. Vibratome sections were obtained from neocortical tissue embedded in 3% low melting agarose using a Leica 1000 vibratome. The sections were 50 µm thick and stored in PBS for further processing. For cryosectioning, neocortical tissue was incubated in a 30% sucrose solution overnight at 4°C, embedded

in Tissue-TEK OCT compound (Sakura Finetek) and stored at −20°C. Cryosections were 10–16 µm thick.

The sections were subjected to antigen retrieval by incubation in 0.01 M sodium citrate (pH 6.0) in a water bath at 70°C for 1 hr. Sections were then incubated with 0.3% Triton X-100 in PBS for 30 min for permeabilization, followed by 0.1 M glycine for 30 min for quenching. Afterwards, sections were incubated overnight at 4°C with primary antibodies in 1x PBS containing 0.2% gelatin, an additional 300 mM NaCl and 0.3% Triton X-100. The sections were incubated with secondary antibodies for 1 hr at room temperature. The following primary antibodies were used in this study: rabbit anti-Sox9 (Sigma, HPA001758, 1:300), rat anti-RFP (ChromoTek, 5F8, 1:500), chicken anti-GFP (Aves labs, GFP-1020, 1:500), rabbit anti-Ki67 (Abcam, ab15580, 1:200), mouse anti-PCNA (Millipore, CBL407, 1:300), goat anti-Sox2 (Santa Cruz, sc-17320, 1:200), rabbit anti-Tbr2 (Abcam, ab23345, 1:200), sheep anti-Tbr2 (R+D Systems, AF6166, 1:500), mouse anti-Olig2 (Millipore, MABN50, 1:200), mouse anti-NeuN (Millipore, MAB377, 1:200), rat anti-phosphohistone H3 (Abcam, ab10543, 1:500), mouse anti-phosphovimentin (Abcam, ab22651, 1:300), mouse anti-Satb2 (Abcam, ab51502, 1:300), rabbit anti-Tbr1 (Abcam, ab31940, 1:200). The secondary antibodies used in this study were donkey- or goat-derived and were coupled to Alexa Fluor 405, 488, 555 and 647 (Life Technologies). Secondary antibody stainings (except when Alexa Fluor 405 was used) were combined with DAPI (Sigma).

### Image acquisition

Images were acquired using Zeiss LSM700 and LSM880 confocal microscopes with 20x, 40x, and 63x objectives. Images scanned in tiles were stitched using ZEN software. All images were visualized and processed using Fiji software (https://fiji.sc/).

### Quantification and statistical analysis

All cell counts were performed using the Fiji software in standardized microscopic fields and calculated in Microsoft Office Excel. Statistics analyses were conducted using Prism (GraphPad Software) and unpaired Student's t-test was used. Sample sizes, the statistical test and significance are indicated in each Figure legend.

### FACS

Dorsolateral cortices were dissected from hemispheres electroporated with control or conditional Sox9 expression plasmids. In each experiment, cortices from 1 to 3 embryos from 2 to 5 mothers were pooled and dissociated into single cell suspension using the MACS Neural Tissue Dissociation kit with papain (Miltenyi Biotec) according to manufacturer's instructions. Dissociated cells were transferred into 5 ml tubes (Falcon) through a 35 µm cell-strainer cap. FACS was performed with a 5-laser-BD FACSAria Fusion (BD Bioscience) and analyzed with FACS Diva software v8.0 (BD Bioscience). First, live cells were identified based on their size and shape and a gate for live cells was set on the SSC/FSC dot-plot. Then a gate was set on FSC-W/FSC-H for sorting. Cells from a non-electroporated brain were used as negative control and fluorescence for RFP+ electroporated cells was detected. Out of the live cells (see above), single dot-plots were created for FSC-H/PE-Texas Red-A (yellow/green laser, 561 nm) and two gates were created for low intensity (low RFP) and high intensity RFP (high RFP) fluorescence. For qPCR experiments, 12,000 cells each from RFP−, low RFP and high RFP populations were sorted into 350 µl of RLT lysis buffer (QIAGEN) with 2 µl of β-mercaptoethanol. For RNA-sequencing experiments, 5,000 cells each from low RFP and high RFP populations were sorted into 350 µl of RLT lysis buffer with 2 µl of β-mercaptoethanol.

### qPCR

Total RNA from sorted cells was extracted using the QIAGEN RNeasy Micro kit according to the manufacturer's protocol. cDNA was synthesized with random hexamers and Superscript III Reverse Transcriptase (Life Technologies, #18080044). qPCR experiments were performed on Mx3000P system (Agilent Technologies) using Absolute qPCR SYBR Green mix (Thermo Scientific). Relative mRNA levels were calculated using the comparative $2^{-\Delta\Delta CT}$ method (*Livak and Schmittgen, 2001*). The housekeeping gene *Gapdh* was used as reference gene. Each experiment was performed in triplicate and with 4–5 biological replicates.

Primers used for qPCR experiments:

Sox9-F: 5'-AGGAAGCTGGCAGACCAGT-3'
Sox9-R: 5'-CTCCTCCACGAAGGGTCTCT-3'
Olig2-F: 5'-CCCCAGGGATGATCTAAGC-3'
Olig2-R: 5'-CAGAGCCAGGTTCTCCTCC-3'
Tbr2-F: 5'-GACCTCCAGGGACAATCTGA-3'
Tbr2-R: 5'-GTGACGGCCTACCAAAACAC-3'
Gapdh-F: 5'-TGAAGCAGGCATCTGAGGG-3'
Gapdh-R: 5'-CGAAGGTGGAAGAGTGGGAG-3'

## RNA sequencing

Total RNA from sorted cells was extracted using the QIAGEN RNeasy Micro kit according to the manufacturer's protocol. cDNA synthesis was performed with SmartScribe reverse transcriptase (Clontech), universal poly-dT primers and template switching oligos. Purified cDNA was amplified with Advantage 2 DNA Polymerase in 12 cycles, and subjected to ultrasonic shearing with Covaris S2. Afterwards, standard Illumina fragment libraries were prepared using NEBnext chemistries (New England Biolabs). Library preparation consisted of fragment end-repair, A-tailing and ligation to indexed Illumina TruSeq adapters, followed by a universally primed PCR amplification of 15 cycles. Then, libraries were purified using XP beads (Beckman Coulter) and quantified by qPCR. Samples were subjected to Illumina 75 bp single end sequencing on an Illumina NextSeq platform.

## Transcriptome analysis

Reads of the same sample on different sequencing lanes were combined and processed by adapter trimming using cutadapt. Alignment of the processed reads to a mouse reference genome (mm10) was performed by STAR. RNA sequencing data were expressed as FPKM (fragments per kilobase of exon per million fragments mapped) values by quantification of the genes from Ensemble release 70 (*Supplementary file 2*). Replicates were clustered using Jensen-Shannon divergence and differential expression analysis was performed using DESeq2. Differential expression analysis was implemented with raw counts as input and differentially expressed genes were identified using a cutoff of q < 0.01 (q refers to a False Discovery Rate (FDR) adjusted p-value). Functional annotation clustering of the differentially expressed genes was performed using DAVID (https://david.ncifcrf.gov/) or Enrichr (http://amp.pharm.mssm.edu/Enrichr/) with default settings and using genes as input. Data visualization was performed using Cummerbund and R (http://www.r-project.org).

Regarding the use of previously published RNA-seq data, differences in mapping strategies between these studies concern the used reference genomes as well as the aligner (*Fietz et al., 2012*): mm9/NCBI37 and TopHat v1.2.0; *Florio et al., 2015*: mm10/GRCm38 and TopHat v2.0.11) whereas both studies quantified genes of the Ensembl release 61 with Cufflinks v0.9.3.

## Organotypic slice culture

Wildtype E14.5 mouse brains were first embedded in 3% low melting point agarose and 250 μm thick sections were obtained using a vibratome. The agarose on the sections was removed and slices were embedded in type Ia collagen (Cellmatrix, Nitta Gelatin) at a concentration of 1.5 mg/ml in DMEM and neutralizing buffer at room temperature, as described in the manufacturer's protocol. Slices in the collagen mix were placed in 35 mm Petri dishes with a 14 mm microwell (MatTek Cooperation). Within a section the two hemispheres were separated, one hemisphere was embedded into type Ia collagen only as control, and the other was embedded into type Ia collagen mixed with either 0.1 mg/ml laminin 211 (Biolamina, human recombinant laminin LN211) or 0.1 mg/ml collagen IV (Abcam, Natural Human Collagen IV protein (FAM) ab123531). Petri dishes with embedded slices were kept at 37℃ for 40 min to allow the collagen mix to set completely. Afterwards 2 ml of slice culture medium was added into each dish and the slices were cultured for 24 hr in a humidified incubation chamber supplied with 40% $O_2$, 5% $CO_2$ and 55% $N_2$.

## Acknowledgements

We thank the Services and Facilities of the Max Planck Institute of Molecular Cell Biology and Genetics for their outstanding technical support, notably Jussi Helppi and his team of the Biomedical

Services (BMS), Jan Peychl and his team of the Light Microscopy Facility, Mihail Sarov and his team of the Genome Editing Facility and Lena Hersemann from the team of Scientific Computing Facility. We would like to thank Robert Lachmann of the Klinik und Poliklinik für Frauenheilkunde und Geburtshilfe, Universitätsklinikum Carl Gustav Carus, for fetal human brain samples, Andreas Dahl and the Deep Sequencing Group of the DFG Research Center for Regenerative Therapies for performing the RNA-sequencing, Katrin Reppe and Anna Pfeffer from the BMS for veterinary assistance with ferret surgeries, and Ilka Reichardt and Dana Olbert from the Genome Editing Facility for their help with Cas9 *in vitro* reactions. We are also grateful to Christiane Haffner for technical assistance, Fong Kuan Wong for kindly providing the control plasmid, Miguel Turrero-Garcia for help with perfusion of ferret kits, Alex Sykes for assistance with molecular cloning, Mareike Albert for help with FACS and RT-PCR, Milos Kostic for advice, Takashi Namba for his constructive comments on the manuscript, and all members of the Huttner group for helpful discussions. AG and MF were members of the International Max Planck Research School for Cell, Developmental and Systems Biology and doctoral students at the Technische Universität Dresden.

## Additional information

### Funding

| Funder | Grant reference number | Author |
|---|---|---|
| Deutsche Forschungsgemeinschaft | SFB 655 | Wieland B Huttner |
| Deutsche Forschungsgemeinschaft | A2 | Wieland B Huttner |
| European Research Council | 250197 | Wieland B Huttner |

The funders had no role in study design, data collection and interpretation, or the decision to submit the work for publication.

### Author contributions

Ayse Güven, Conceptualization, Formal analysis, Investigation, Visualization, Methodology, Writing - original draft, Writing - review and editing, Conceptualization, Formal analysis, Investigation, Visualization, Methodology, Writing-original draft, Writing-review and editing, Performed in utero electroporations in mice, cell sorting, RNA isolation, RT-PCR and gene expression analyses; Nereo Kalebic, Formal analysis, Investigation, Visualization, Methodology, Writing - review and editing, Formal analysis, Investigation, Visualization, Methodology and Writing related to the Sox9 KO and in vivo experiments in ferret; Katherine R Long, Methodology, Writing - original draft, Methodology, Assisted organotypic slice cultures, Writing - input for original draft; Marta Florio, Methodology, Methodology, Analysis and interpretation of the RNA-sequencing data and GO term analyses, Assisted human tissue dissection, Assisted FACS; Samir Vaid, Methodology, Methodology, Performed in utero electroporations in mice-review; Holger Brandl, Methodology, Analysis of the RNA-sequencing data; Denise Stenzel, Conceptualization, Supervision, Investigation, Methodology, Conceptualization, Investigation, Methodology, Provided supervision to Ayse Güven; Wieland B Huttner, Conceptualization, Supervision, Funding acquisition, Writing - original draft, Project administration, Writing - review and editing, Conceptualization, Interpretation of data, Supervision, Funding acquisition, Project administration, Writing-original draft, Writing-review and editing

### Author ORCIDs

Nereo Kalebic (ID) https://orcid.org/0000-0002-8445-2906
Katherine R Long (ID) http://orcid.org/0000-0003-0660-2486
Holger Brandl (ID) http://orcid.org/0000-0003-1911-8570
Wieland B Huttner (ID) https://orcid.org/0000-0003-4143-7201

## Ethics

Human subjects: Human fetal neocortical tissue was provided by the Klinik und Poliklinik für Frauenheilkunde und Geburtshilfe, Universitätsklinikum Carl Gustav Carus, involving elective pregnancy terminations with informed written maternal consents and approval by the local University Hospital Ethical Committees. Human fetal neocortical tissue was also obtained from Novogenix Laboratories (Torrance, CA).

Animal experimentation: All animal experiments in this study were conducted according to the German animal welfare legislation. All experiments utilizing mice and ferrets were conducted in agreement with the German Animal Welfare Legislation after approval by the Landesdirektion Sachsen (licence TVV 05/2015 and TVV 21/2017).

## Decision letter and Author response

Decision letter https://doi.org/10.7554/eLife.49808.sa1
Author response https://doi.org/10.7554/eLife.49808.sa2

# Additional files

## Supplementary files

• Supplementary file 1. Lists of genes whose expression is up- or downregulated 48 hr after conditional Sox9 expression. Tab-1 (Sox9_up_PC): List of genes that are upregulated upon conditional Sox9 expression. Tab-2 (Sox9_down_PC): List of genes that are downregulated upon conditional Sox9 expression. Columns in each tab denote the following information consecutively: A: Ensemble gene ID, B: Gene name, C: Description, D: Average count for Sox9 sample, E: Average count for control sample, G: Logarithmic fold change (FC) of counts between Sox9 and control samples, J: p-value, K: adjusted p-value, W: Average FPKM value for control sample, X: Average FPKM value for Sox9 sample, Y: Fold change (FC) between average FPKM values of Sox9 and control samples, Z: Logarithmic fold change (FC) between average FPKM values of Sox9 and control samples.

• Supplementary file 2. Datasets of RNA-sequencing and gene ontology (GO) term enrichment analysis of Sox9 and control samples. Tab-1 (FPKM): FPKM values for each replicate of Sox9 and control samples. Tab-2 (GO_BP): Gene ontology (GO) term enrichment analysis for biological process (BP). Tab-3 (GO_CC): Gene ontology (GO) term enrichment analysis for cellular component. Tab-4 (KEGG): KEGG pathway analysis.

• Supplementary file 3. List of ECM-related genes that were identified by GO term and KEGG pathway analyses. Tab-1 (ECM-related genes): List of all genes that were enriched for ECM related terms (*Figure 5C,D and E*) in GO term and KEGG pathway analyses.Columns in each tab denote the following information consecutively: A: Ensemble gene ID, B: Gene name, C: Description, D: Average count for Sox9 sample, E: Average count for control sample, G: Logarithmic fold change (FC) of counts between Sox9 and control samples, J: p-value, K: adjusted p-value, W: Average FPKM value for control sample, X: Average FPKM value for Sox9 sample, Y: Fold change (FC) between average FPKM values of Sox9 and control samples, Z: Logarithmic fold change (FC) between average FPKM values of Sox9 and control samples.

• Supplementary file 4. Key Resources Table.

• Transparent reporting form

## Data availability

Sequencing data have been deposited in GEO under accession code GSE134162.

The following dataset was generated:

| Author(s) | Year | Dataset title | Dataset URL | Database and Identifier |
|---|---|---|---|---|
| Guven A, Florio M, Brandl H, Huttner WB | 2019 | Extracellular matrix-inducing Sox9 orchestrates basal progenitor proliferation and gliogenesis in developing neocortex | https://www.ncbi.nlm.nih.gov/geo/query/acc.cgi?acc=GSE134162 | NCBI Gene Expression Omnibus, GSE134162 |

The following previously published datasets were used:

| Author(s) | Year | Dataset title | Dataset URL | Database and Identifier |
|---|---|---|---|---|
| Fietz SA, Huttner WB, Pääbo S | 2012 | Transcriptomes of germinal zones of human and mouse fetal neocortex suggest a role of extracellular matrix in progenitor self-renewal | https://www.ncbi.nlm.nih.gov/geo/query/acc.cgi?acc=GSE38805 | NCBI Gene Expression Omnibus, GSE38805 |
| Florio M, Albert M, Huttner WB | 2015 | Human-specific gene ARHGAP11B promotes basal progenitor amplification and neocortex expansion | https://www.ncbi.nlm.nih.gov/geo/query/acc.cgi?acc=GSE65000 | NCBI Gene Expression Omnibus, GSE65000 |

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
