## [Decision Letter]

**Acceptance summary:**

In this manuscript, the authors identified a pool of SOX9+ bRGs in the subventricular zone of human and ferret cortex but absent in mouse sub ventricular zone. Forced SOX9 expression in mouse basal progenitors promoted cell proliferation and increased both neurogenesis and gliogenesis, consistent with a role of SOX9 expression in neocortical expansion. SOX9 forced expression in mouse neural precursors promoted basal progenitor proliferation both cell autonomously and non-cell autonomously upregulated expression of ECM genes. From this the authors tested the ECM candidate Laminin 211, which promoted proliferation of basal progenitors.

**Decision letter after peer review:**

Thank you for submitting your article "Extracellular matrix-inducing Sox9 orchestrates basal progenitor proliferation and gliogenesis in developing neocortex" for consideration by *eLife*. Your article has been reviewed by three peer reviewers, and the evaluation has been overseen by Marianne Bronner as the Senior/Reviewing Editor. The following individual involved in review of your submission has agreed to reveal their identity: Silvia Nicolis (Reviewer #3).

The reviewers have discussed the reviews with one another and the Reviewing Editor has drafted this decision to help you prepare a revised submission.

Essential revisions:

The reviewers were enthusiastic about the paper but felt that some major revisions were required. In particular:

1) Why was laminin 211 selected for study?

2) How does laminin 211 work to increase proliferation?

3) Can you demonstrate that Sox9 loss of function in gyrated cortex impacts BP proliferation or fate?

Many of the other changes should be relatively straight-forward. Please see the attached full reviews to guide you in revising the paper.

Reviewer #1:

The is a very interesting manuscript on a very important and hot topic of general interest to the field of neuroscience and evolution. The manuscript is generally well written, the rationale is very clear and easy to follow, the datasets and figures are generally very clear, and for the most part conclusions are well supported by the experimental results. However, there are a few points that must be considered before this manuscript is ready for publication, which the authors are perfectly capable of addressing.

1) There is no justification as to choice of ferret developmental stages for their experiments, and this needs clarifying. Likewise, there is no justification as to why several different stages are analyzed in each case, when they are very similar and in fact considered equivalent (i.e. P2 and P3 in Figure 2). Most importantly, the authors only analyze 1 animal or embryo of each age, which renders it of limited use for statistical analysis. This is true in all analyses on ferret and human tissue throughout the study. At least for ferret, 3 animals must be analyzed per stage and variable studied.

2) In Figure 3F, H, the authors count number of progenitor cells expressing the reporter protein per field of analysis. This is wrong because these experiments were done by *in utero* electroporation, and hence the abundance of cells expressing the reporter gene depends on the abundance of electroporated cells, which may vary quite significantly from embryo to embryo. These results should be eliminated from the study altogether.

3) The images shown in Figure 3N, O are useless for the purpose of demonstrating/illustrating the degree of marker co-expression. High magnification images must be provided, as in Figure 3C', D', J' and M.

4) The authors conclude that "Sox9-induced increase in the proliferation of mouse BPs pertained primarily to bIPs", based on the absence of basal processes in phosphovimentin+ cells. While this interpretation is correct, one should never make strong statements based on the absence of evidence, in this case absence of a basal process. The authors must confirm this finding by analyzing Pax6 expression in these BPs. In mouse, bIPs are positive for Tbr2 and negative for Pax6, whereas bRGs (even if scarce in normal embryos) are Pax6+ and Tbr2-.

5) There is a typo where indicating the p value for significance: "q<0.01". Q should be changed by a P.

6) In Figure 6, when the authors study whether the effects of *Sox9* are cell-autonomous or not, they evaluate cells positive for NeuN (pan-neuronal marker) in the mouse Cortical Plate, where newborn neurons finish migration and start differentiation. Surprisingly, they find that in control embryos only 50% of CP cells are positive for NeuN, and 75% in *Sox9*-expressing embryos. What are the cells negative for NeuN? It is difficult to imagine such a high abundance of endothelial or other cell types, especially so early in development, prior to gliogenesis.

7) In the subsection “Conditional Sox9 expression induces cell non-autonomous BP proliferation resulting in increased neuron production, especially of upper-layer neurons”, the authors conclude that Sox9 expression in BPs results in increased neuron production from BPs in a cell non-autonomous manner. They reach this conclusion after showing that there is only an increase in neurons negative for the reporter protein (RFP), but not in RFP+ neurons. This conclusion seems correct based on the results in RFP- cells, but then why are RFP+ cells not affected? Why Sox9+ cells exert this influence on neighboring cells only if those are Sox9-, but not on those that are Sox9+? Shouldn't the non cell-autonomous effect be indifferent to Sox9 expression? Is there a mechanism for this effect to be restricted to Sox9- (RFP-) cells only? This needs a clarifying explanation.

8) According to the results in Figures 7 and 8, the authors find that the non cell-autonomous effect of Sox9-expressing cells only applies to the production of Satb2+ neurons, but then the influence on the production of Tbr1+ neurons is cell-autonomous. This is problematic and the authors offer no explanation, but rather stick to the non cell-autonomous model. Additional arguments and/or data must be provided to clarify this point of conflict.

9) In the statistical analysis of data in Figure 7B, did the authors perform multiple comparisons before t-test?

10) In Figure 7C, D, it is impossible to see co-expression of markers with the images provided. High magnifications must be added to illustrate this result, as already mentioned above for other figures. Related to these results, the authors must quantify density of Satb2+ neurons/field, and Satb2+/RFP- cells/field, as they do previously, to demonstrate increased abundance of this neuron type. Otherwise, their counts only show no difference in a cell-autonomous manner.

11) In Figure 8A, B, are the values of transcript abundance statistically different? Some of it is mentioned in the text, but needs to be indicated in the figure. Laminin α2 is said to not be expressed at significantly different levels in the RNAseq data, while Sox9 was very different. This contradicts the conclusion that the effects of Sox9 on cortical progenitors are via increased expression of laminin α2. On this regard, the authors' conclusion is quite weak and circumstantial, as there is no functional data demonstrating a link between the phenotypes produced by Sox9 overexpression and laminin α2 treatment, even if these are similar.

12) Images in Figure 8C must be replaced by better examples and cropped properly. In their current format there is excessive background, the apical surface of the VZ is barely visible (and thus also apical mitoses), and the field of view of the images is so excessively narrow that the control slice shows a complete absence of basal mitoses.

Reviewer #2:

While their findings are of potential interest, the manuscript has several weaknesses.

1) Loss-of-function evidence to support a role of SOX9 in neocortical expansion in gyrated mammalian brain is lacking. Knockdown or knockout SOX9 in ferret (or possibly human organoid) bRGs would be necessary to demonstrate the conclusions.

2) In their study, the authors forced SOX9 expression through *in utero* electroporation DNA construct into tamoxifen-treated E13.5 embryos of the *Tis21*-CreER^T2^ mouse lines. Why wouldn't this lead to expression of SOX9 in derived neurons and glial cells as well as the BPs? (see Figure 3D). This may lead to artifacts, if more glia are generated between the dose of tamoxifen and harvest. i.e. What were the distributions of cell types that were harvested?

3) Which specific ECM genes were upregulated after forced SOX9 expression? From the RNA-seq result in Table 1, only LAMA3 and A4 are upregulated, while the genes expressing Laminin β and γ chains are not listed. Was LAM211 components increased upon SOX9 misexpression? Justification for LAM211 was missing.

4) Previous studies showed SOX9 functions in neural stem cell proliferation and gliogenesis (Kang et al., 2012; Scott et al., 2010), partially weakening the novelty of this study, but the connection to evolution is novel.

5) Readers will still be confused on the overall model. Does LAM211 produced by bRGs influencing the bRGs that surround them (paracrine) or some other mechanisms? Is this effect spatially restricted?

Reviewer #3:

The paper is very interesting, in particular regarding a possible non-cell-autonomous action of Sox9 through extracellular matrix molecules, and the experiments are well conducted. However, some clarifications would be important.

1) The authors show that, interestingly, laminin 211 stimulates cell proliferation in slices. They also show that Sox9 stimulates the expression of Lama2, encoding a laminin. Would the expression of Lama2 by *in utero* electroporation have the same (or similar) effect of the expression of Sox9 on cell proliferation? (and perhaps gliogenesis)? This would much strengthen the evidence that laminins/ECM are important mediators of the function of Sox9 on progenitor cell proliferation (and perhaps gliogenesis), as interestingly suggested by the data. Even if the experiment did not work, the authors could still mention it and discuss it.

2) The proliferating cells in the SVZ of ferret and human express Sox9; however, they also coexpress other transcription factors, e.g. Sox2 (as also shown by the authors), typical of proliferating progenitors in general (in fact, both are also expressed in the VZ). So, one may argue that Sox2 (or another coexpressed Sox factor, or other factor) could also be (together with Sox9) a candidate for these functions of the SVZ cells, as part of a genetic program of proliferating cells which has been recruited to SVZ cells through evolution. These possibilities should be discussed more in depth. Also, the effects (or not) of the Sox9 knockouts on cell proliferation should be mentioned. Overall, although the observed effects of ectopic Sox9 expression are very interesting, they are not in my opinion sufficient to qualify Sox9 as a "master regulator to promote neocortical expansion" (Abstract) nor to state (Discussion) that "The present study establishes Sox9 as a master regulator of BP proliferation and cell fate"; I would moderate these statements a little.

[Editors' note: further revisions were suggested prior to acceptance, as described below.]

Thank you for submitting your article "Extracellular matrix-inducing Sox9 orchestrates basal progenitor proliferation and gliogenesis in developing neocortex" for consideration by *eLife*. Your article has been reviewed by three peer reviewers, and the evaluation has been overseen by Marianne Bronner as the Senior/Reviewing Editor. The following individual involved in review of your submission has agreed to reveal their identity: Joseph G Gleeson (Reviewer #2).

The reviewers have discussed the reviews with one another. While the paper is much improved, there are remaining concerns, detailed below that need to be addressed before a final decision can be made.

Reviewer #1:

The manuscript has been very nicely revised following the comments and concerns on its initial version. Unfortunately, there are still a handful of points that were not addressed properly:

Point 4:

It is very nice that the authors performed this analysis. However, it is very surprising and strange that in control electroporated cortex of mouse embryos more than 70% of RFP+pVim+ cells in SVZ-IZ are Pax6+ (less than 30% are Pax6-; new Figure 4—figure supplement 3). A large body of literature, including studies from the Huttner lab, demonstrates that the vast majority of mouse cortical BPs are bIPs, hence most mitotic cells in SVZ-IZ are Pax6-.

Point 9:

This is incorrect. Multiple comparisons tests must be performed (i.e. ANOVA) when comparing value distributions, and then only if this reveals a statistical difference one must perform t-tests bin by bin.

Point 11:

The argument of this reviewer is well understood, but the issue here is that of expression levels: direct overexpression of laminin 211 increases BP proliferation, similar to overexpression of Sox9. But is the increased expression level of Laminin α2 caused by Sox9 overexpression of sufficient magnitude to really mediate this effect? In other words, the authors' conclusions would be much stronger if they could rescue the effects of Sox9 overexpression by blockade of its effect on Laminin 211 expression levels.

Point 12:

The argument about background is acceptable, and the new high magnifications are nice, but the comment that apical mitoses are only partly visible was completely ignored. Although there is no change in apical mitosis abundance, these histological results must still be shown and demonstrated properly. The current format (same as in the previous manuscript version) is unacceptable.

Reviewer #3:

The paper has been improved by the new experiments of Sox9 CRISPR-Cas9-mediated Sox9 KO in the ferret, which are very interesting. However, I maintain a major concern about the overemphasis given to Sox9 as a "key regulator" of cortical size increase during evolution, which is not, in my opinion, supported by the data (see below). The findings remain interesting, but overemphasis in interpretation is misleading, and should be avoided.

Sox9 activation in mouse BP leads to a statistically significant, but very moderate in absolute terms, increase of BP proliferation, starting from a very low background of spontaneous (i.e. not induced by transgene) BP proliferation. The authors should take this point into serious consideration in the Discussion and throughout the paper, and substantially attenuate their proposal that Sox9 expression in BP is a key element explaining the expansion of the cortex in evolution.

The important effects shown by the KO in the ferret (though see point 2 below, to be clarified), which are certainly a worthy addition to the paper, should be reconciled with the relative ineffectiveness of Sox9 overexpression in mouse; perhaps *Sox9* expression in BP in the ferret is accompanied by expression of other important factors, and it is the synergy of Sox9 with these other factors that matters in the expansion of the cortex.

In the new Figure 3 (Sox9 ko in ferret), the PCNA -mCherry/mCherry -positive cells within the VZ are reported to be numerically similar between Sox9-mutant and control ferrets (histograms in Figure 3B), however the intensity of PCNA staining in the Sox9-mutant VZ (Figure 3A) is much lower than in control. This makes one fear that the 50% reduction of PCNA-positive cells reported for the mutant OSVZ (Figure 4B) may be contributed to by a lower efficiency of the staining. This should be clarified, the image shown and the data in the histogram should be consistent.

---

## [Author Response]

Essential revisions:The reviewers were enthusiastic about the paper but felt that some major revisions were required. In particular:1) Why was laminin 211 selected for study?

As explained in the revised manuscript, laminin 211 (laminin α2, β1, ɣ1) was chosen as a representative member of the laminin family of proteins, because laminin α2, laminin β1 and laminin ɣ1 have previously been shown to be expressed in the embryonic mouse and fetal human VZ and SVZ, notably the human OSVZ (Lathia et al., 2007; Fietz et al., 2012; Florio et al., 2015), because laminin α2 has recently been shown to promote progenitor proliferation in the developing brain (Ahmed et al., 2019), and because laminin α2 was found to be increased upon conditional Sox9 expression.

2) How does laminin 211 work to increase proliferation?

The group of Charles Ffrench-Constant (Ahmed et al., 2019) recently showed that laminin 211 promotes midbrain progenitor proliferation via integrins. In exploring a role of extracellular matrix (ECM) components on progenitor proliferation in the developing brain, our group previously found that activation of the ECM receptor integrin αvβ3 on basal progenitors in embryonic mouse neocortex promotes their expansion (Stenzel et al., 2014). All these data are consistent with the notion that laminin 211 increases basal progenitor proliferation via integrin signaling, as is discussed in the revised manuscript.

3) Can you demonstrate that Sox9 loss of function in gyrated cortex impacts BP proliferation or fate?

Yes, the new data on the CRISPR/Cas9-mediated knockout of Sox9 in embryonic ferret neocortex (new Figure 3) demonstrate that the Sox9 loss of function in this gyrated cortex results in a substantial reduction of BP proliferation.

Many of the other changes should be relatively straight-forward. Please see the attached full reviews to guide you in revising the paper.

We have addressed all points raised by the reviewers, as is described in our point-by-point response below.

Reviewer #1:[…] There are a few points that must be considered before this manuscript is ready for publication, which the authors are perfectly capable of addressing.1) There is no justification as to choice of ferret developmental stages for their experiments, and this needs clarifying. Likewise, there is no justification as to why several different stages are analyzed in each case, when they are very similar and in fact considered equivalent (i.e. P2 and P3 in Figure 2). Most importantly, the authors only analyze 1 animal or embryo of each age, which renders it of limited use for statistical analysis. This is true in all analyses on ferret and human tissue throughout the study. At least for ferret, 3 animals must be analyzed per stage and variable studied.

As requested by the reviewer, we have now added a clarification regarding the choice of the developmental stages for ferret. Specifically, we chose the developmental stages of the ferret based on: (i) availability of the tissue in order to reduce animal use, (ii) the time of the peak of neurogenesis (Kalebic et al., 2018).

Also in line with the reviewer's comment regarding the analysis of only 1 ferret each – albeit for two equivalent developmental stages –, we have now pooled equivalent developmental stages for ferret (e.g., P2 and P3 in Figure 2) and show the mean of these data, with the two individual values indicated (Figure 2, revised panels B and C). The same type of revision has been carried out for E40-P1 (Figure 5—figure supplement 1, revised panels B and C). The text has been revised accordingly.

For the new data on the CRISPR/Cas9-mediated KO of Sox9 in developing ferret neocortex (new Figure 3), data are the mean of 4 embryos.

2) In Figure 3F, H, the authors count number of progenitor cells expressing the reporter protein per field of analysis. This is wrong because these experiments were done by *in utero* electroporation, and hence the abundance of cells expressing the reporter gene depends on the abundance of electroporated cells, which may vary quite significantly from embryo to embryo. These results should be eliminated from the study altogether.

In our hands, the abundance of electroporated cells does not vary that much from embryo to embryo, as can also be deduced from the similarity of the results obtained when expressing the data per RFP+ cells (previous Figure 3, panels E and G) vs. per field (previous Figure 3, panels F and H). Nonetheless, as requested by the reviewer, we have eliminated the per field data shown in the previous Figure 3 (now Figure 4), panels F and H, and likewise those shown in the previous Figure 3, panel L, and in the previous Figure 4 (now Figure 5), panel D, from the study and revised the figure legends and the text accordingly.

3) The images shown in Figure 3N, O are useless for the purpose of demonstrating/illustrating the degree of marker co-expression. High magnification images must be provided, as in Figure 3C', D', J' and M.

As requested by the reviewer, we have now added high magnification images for the cell cycle re-entry data (previous Figure 3, panels N and O, now Figure 4, panels K, K’, L, L’).

4) The authors conclude that "Sox9-induced increase in the proliferation of mouse BPs pertained primarily to bIPs", based on the absence of basal processes in phosphovimentin+ cells. While this interpretation is correct, one should never make strong statements based on the absence of evidence, in this case absence of a basal process. The authors must confirm this finding by analyzing Pax6 expression in these BPs. In mouse, bIPs are positive for Tbr2 and negative for Pax6, whereas bRGs (even if scarce in normal embryos) are Pax6+ and Tbr2-.

As requested by the reviewer, we have added an analysis of Pax6 expression in BPs (new Figure 4—figure supplement 3). This shows, as anticipated by the reviewer, that conditional Sox9 expression in embryonic mouse neocortex indeed increased the proportion of the targeted cell-derived (RFP+) mitotic (pVim+) BPs that were Pax6-negative, i.e. bIPs.

5) There is a typo where indicating the p value for significance: "q<0.01". Q should be changed by a P.

The q in "q<0.01" was not a typo. The term 'q-value' is a valid expression and refers to an FDR adjusted p-value. This has now been clarified in the revised Materials and methods.

6) In Figure 6, when the authors study whether the effects of Sox9 are cell-autonomous or not, they evaluate cells positive for NeuN (pan-neuronal marker) in the mouse Cortical Plate, where newborn neurons finish migration and start differentiation. Surprisingly, they find that in control embryos only 50% of CP cells are positive for NeuN, and 75% in Sox9-expressing embryos. What are the cells negative for NeuN? It is difficult to imagine such a high abundance of endothelial or other cell types, especially so early in development, prior to gliogenesis.

In answer to the reviewer's question, we believe that most of the NeuN-negative cells are neurons whose maturation process is not yet complete and has not yet resulted in NeuN expression to a level detectable by immunohistochemistry. We agree with the reviewer that it is unlikely that the NeuN-negative cells are non-neuronal cells. This has now been clarified in the revised text.

7) In the subsection “Conditional Sox9 expression induces cell non-autonomous BP proliferation resulting in increased neuron production, especially of upper-layer neurons”, the authors conclude that Sox9 expression in BPs results in increased neuron production from BPs in a cell non-autonomous manner. They reach this conclusion after showing that there is only an increase in neurons negative for the reporter protein (RFP), but not in RFP+ neurons. This conclusion seems correct based on the results in RFP- cells, but then why are RFP+ cells not affected? Why Sox9+ cells exert this influence on neighboring cells only if those are Sox9-, but not on those that are Sox9+? Shouldn't the non cell-autonomous effect be indifferent to Sox9 expression? Is there a mechanism for this effect to be restricted to Sox9- (RFP-) cells only? This needs a clarifying explanation.

We believe that the most likely explanation is related to the relative proportion of cells that are NeuN-positive. As shown in panel J of Figure 7 (previous Figure 6), the proportion of RFP+ cells that are NeuN+ is already almost 80% in the control, so to detect a statistically significant increase in this proportion upon conditional Sox9 expression will be difficult. In contrast, as shown in panel K of Figure 7, the proportion of nuclei that are RFP– and NeuN+ is only around 30% in the control, so one can more readily detect a significant increase in this proportion upon conditional Sox9 expression. This has now been clarified in the revised text.

In addition, we show that a proportion of the RFP+ cells switch to gliogenesis upon conditional Sox9 expression (panel F of revised Figure 5 (previous Figure 4)), which reduces the probability of detecting an increase in the proportion of RFP+ cells that are NeuN+.

8) According to the results in Figures 7 and 8, the authors find that the non cell-autonomous effect of Sox9-expressing cells only applies to the production of Satb2+ neurons, but then the influence on the production of Tbr1+ neurons is cell-autonomous. This is problematic and the authors offer no explanation, but rather stick to the non cell-autonomous model. Additional arguments and/or data must be provided to clarify this point of conflict.

Again, we believe that the most likely explanation is related to the relative proportion of cells that are Satb2+ or Tbr1+, and whether there is an increase or a decrease in this proportion. Specifically, as shown in panel H of revised Figure 8 (previous Figure 7), the proportion of RFP+ cells that are Satb2+ is already >70% in the control, and although there is an increase in this proportion to almost 90% upon conditional Sox9 expression, this increase is not statistically significant, which presumably reflects the already high starting value of the control. In contrast, as shown in panel G of revised Figure 8, the proportion of RFP+ cells that are Tbr1+ is only around 30% in the control, and this proportion is decreased upon conditional Sox9 expression, two parameters that contribute to rendering this effect of Sox9 expression statistically significant. This has now been clarified in the revised text.

9) In the statistical analysis of data in Figure 7B, did the authors perform multiple comparisons before t-test?

We did not perform multiple comparisons in the statistical analysis of the data shown in the previous Figure 7B (now Figure 8B). For each bin (of equal size), we performed a separate t-test, comparing the percentage of RFP-positive cells in the respective bin of the CP 5 days after electroporation with control construct with the respective bin of the CP 5 days after electroporation with conditional Sox9 expression construct. We have now added this information to the legend of Figure 8.

10) In Figure 7C, D, it is impossible to see co-expression of markers with the images provided. High magnifications must be added to illustrate this result, as already mentioned above for other figures. Related to these results, the authors must quantify density of Satb2+ neurons/field, and Satb2+/RFP- cells/field, as they do previously, to demonstrate increased abundance of this neuron type. Otherwise, their counts only show no difference in a cell-autonomous manner.

As requested by the reviewer, we have now added high magnifications of the images shown in the revised Figure 8 (previous Figure 7), panels C-F.

Furthermore, as requested, we have quantified the number of Satb2-positive neurons per field (revised Figure 8 (previous Figure 7), new panel J) and of Satb2-positive, RFP-negative neurons per field (revised Figure 8 (previous Figure 7), new panel I). These quantifications show that the number of Satb2-positive neurons increase in a cell non-autonomous manner upon conditional Sox9 expression, corroborating our previous findings.

11) In Figure 8A, B, are the values of transcript abundance statistically different? Some of it is mentioned in the text, but needs to be indicated in the figure. Laminin α2 is said to not be expressed at significantly different levels in the RNAseq data, while Sox9 was very different. This contradicts the conclusion that the effects of Sox9 on cortical progenitors are via increased expression of laminin α2. On this regard, the authors' conclusion is quite weak and circumstantial, as there is no functional data demonstrating a link between the phenotypes produced by Sox9 overexpression and laminin α2 treatment, even if these are similar.

As requested by the reviewer, we now indicate the statistical significance values in Figure 9A, D (previous Figure 8A, B) and the corresponding figure legend.

Regarding the relationship between laminin α2 and Sox9, there appears to be a misunderstanding. We do not say that "Laminin α2 is not expressed at significantly different levels in the RNAseq data"– in fact it is, as shown in Figure 9A – but that "the laminin α2 mRNA was not among the list of the 870 upregulated genes". The reason for this is that genes with low mRNA FPKM values, like laminin α2, were excluded from the analysis of the genes with increased expression upon conditional Sox9 expression. Importantly, we do not mean to imply that the various effects of Sox9 on cortical progenitors are mediated exclusively via increased expression of laminin α2. However, in light of our findings that (i) conditional Sox9 expression does increase laminin α2 mRNA levels (Figure 9A) and (ii) laminin 211 increases basal progenitor proliferation, we find it likely that increased laminin expression contributes to the increase in basal progenitor proliferation observed upon conditional Sox9 expression. We have clarified this issue in the revised manuscript.

12) Images in Figure 8C must be replaced by better examples and cropped properly. In their current format there is excessive background, the apical surface of the VZ is barely visible (and thus also apical mitoses), and the field of view of the images is so excessively narrow that the control slice shows a complete absence of basal mitoses.

The background reflects the fact that the slices were embedded in collagen, and so there is little we can do about this. To present the basal mitoses in a more convincing manner, we have now added high magnification images (Figure 9 (previous Figure 8), new panel E'), which clearly show abventricular mitoses.

Reviewer #2:While their findings are of potential interest, the manuscript has several weaknesses.1) Loss-of-function evidence to support a role of SOX9 in neocortical expansion in gyrated mammalian brain is lacking. Knockdown or knockout SOX9 in ferret (or possibly human organoid) bRGs would be necessary to demonstrate the conclusions.

We thank the reviewer for this very constructive comment. As suggested by the reviewer, we have now performed CRISPR/Cas9-mediated knock-out of Sox9 in embryonic ferret neocortex and observed the following upon ablation of Sox9: (i) a marked reduction of proliferating, PCNA-positive BPs in the ISVZ and OSVZ (new Figure 3, panel B); (ii) a decrease in mitoses in the germinal zones as revealed by pVim immunofluorescence, which was particularly striking for the ISVZ and OSVZ, indicative of a massive reduction of mitotic BPs (new Figure 3, panel C); (iii) a strong reduction of Sox2-positive neural progenitor cells in the ISVZ and OSVZ (new Figure 3, panel D), which pertained mostly to bRG-like cells bearing radial processes (new Figure 3, panel E). Taken together, these new data provide the requested loss-of-function evidence in support of a role of Sox9 in neocortical expansion in a gyrated mammalian brain.

2) In their study, the authors forced SOX9 expression through *in utero* electroporation DNA construct into tamoxifen-treated E13.5 embryos of the Tis21-CreER^T2^ mouse lines. Why wouldn't this lead to expression of SOX9 in derived neurons and glial cells as well as the BPs? (see Figure 3D). This may lead to artifacts, if more glia are generated between the dose of tamoxifen and harvest. i.e. What were the distributions of cell types that were harvested?

As anticipated by the reviewer, our approach of conditional Sox9 expression in mouse BPs will presumably result in some Sox9 expression in the postmitotic neurons and glial cells derived from these BPs (see, for example, Figure 4D, which is two days after *in utero* electroporation). However, as our study in focused on the effects of conditional Sox9 expression on progenitor cells in the SVZ, we do not see why maintenance of Sox9 expression in the postmitotic neurons and glial cells derived therefrom should pose a problem. Specifically, in answer of the reviewer's question, we have compared the distribution of cells between neocortices electroporated with control and conditional Sox9 construct, and observed no significant difference. These data have now been added to Figure 4—figure supplement 2, new panels C-E.

3) Which specific ECM genes were upregulated after forced SOX9 expression? From the RNA-seq result in Table 1, only LAMA3 and A4 are upregulated, while the genes expressing Laminin β and γ chains are not listed. Was LAM211 components increased upon SOX9 misexpression? Justification for LAM211 was missing.

The reviewer is correct in stating that in Supplementary file 1, only the laminin α3 and the laminin α4 mRNAs, but not laminin β and γ chain mRNAs, are listed as being upregulated upon conditional Sox9 expression. The reason why the laminin α2 mRNA was not listed in that table is that because of its low mRNA FPKM value, it was not included in the analysis of the genes with increased expression upon conditional Sox9 expression. However, as shown in the previous Figure 8A (now Figure 9A), the laminin α2 mRNA is significantly increased upon conditional Sox9 expression.

In light of these data, we felt it was legitimate to focus on laminin 211. The laminin β and γ chains required to form laminin 211 are lamb1 and lamc1, both of which are expressed in the germinal zones / progenitor cells of embryonic mouse neocortex (Fietz et al., 2012, Florio et al., 2015). Although these mRNAs were not differentially expressed upon conditional Sox9 expression (data now shown in panels B and C of Figure 9) and therefore not included in Supplementary file 1, their endogenous expression levels would be sufficient to form laminin 211 together with the differentially expressed lama2 α chain. This has now been clarified in the revised text.

In addition, in the new Supplementary file 3, we now list those of the genes up-regulated upon conditional Sox9 expression that showed the highest scores in the GO term enrichment and KEGG pathway analyses (previous Figure 5, now Figure 6), i.e. that are ECM-related.

4) Previous studies showed SOX9 functions in neural stem cell proliferation and gliogenesis (Kang et al., 2012; Scott et al., 2010), partially weakening the novelty of this study, but the connection to evolution is novel.

The reviewer raises a fair point. One of the studies pointed out by the reviewer, from the Briscoe lab, indeed shows the influence of Sox9 on neural stem cell proliferation at E10.5 in mouse, at a stage when the neuroepithelial cells start generating aRGs. This demonstration of a role of Sox9 in apical progenitor proliferation clearly constitutes an important contribution. However, as pointed out by the reviewer, our study extends the previous work by showing a novel role of Sox9 in basal progenitor proliferation and fate choice that is essential and exclusive to gyrencephalic species such as ferret and human. The study from the Briscoe lab is now cited in the revised manuscript.

The other study pointed out by the reviewer (Kang et al., 2012), already cited and discussed in the previous version of our manuscript, focuses on the role of Sox9 in developing spinal cord and shows an elegant mechanism for Sox9 to exert its effects on gliogenic fate choice by partnering with another transcription factor, NFIA. Again extending this work, our study shows a significant role for Sox9 in the initiation of gliogenesis in the neocortex, which was not shown before.

5) Readers will still be confused on the overall model. Does LAM211 produced by bRGs influencing the bRGs that surround them (paracrine) or some other mechanisms? Is this effect spatially restricted?

To clarify matters, the revised manuscript now includes a model (Supplementary file 1) that summarizes the effects of conditional Sox9 expression on basal progenitor proliferation, fate choice, and neuron output from basal progenitors. This model illustrates the proposed roles of local ECM production. We do not know the spatial range of the effects of the locally produced ECM.

As to the underlying mechanism, the laminin α chain is known to bind integrins (Loulier et al., 2009), which in turn have been shown to promote bRG and bIP proliferation through the pERK pathway (Ahmed et al., 2019; Fietz et al., 2010; Hall et al., 2008; Radakovitz et al., 2009; Stenzel et al., 2014, Long et al., 2016).

Reviewer #3:The paper is very interesting, in particular regarding a possible non-cell-autonomous action of Sox9 through extracellular matrix molecules, and the experiments are well conducted. However, some clarifications would be important.1) The authors show that, interestingly, laminin 211 stimulates cell proliferation in slices. They also show that Sox9 stimulates the expression of Lama2, encoding a laminin. Would the expression of Lama2 by *in utero* electroporation have the same (or similar) effect of the expression of Sox9 on cell proliferation? (and perhaps gliogenesis)? This would much strengthen the evidence that laminins/ECM are important mediators of the function of Sox9 on progenitor cell proliferation (and perhaps gliogenesis), as interestingly suggested by the data. Even if the experiment did not work, the authors could still mention it and discuss it.

The suggestion by the reviewer to study the effects of Lama2 expression by *in utero* electroporation is very interesting. We have seriously thought about this experiment, but also needed to consider the following. Laminins are large heterotrimeric multidomain proteins and are functional in form of trimers composed of α, β and γ chains. Lama2 could form a trimer with any laminin β and γ chain. For Lama2 expression by *in utero* electroporation to mimic the effects of laminin 211 in slice culture, Lama2 would need to form a complete laminin trimer with Lamb1 and Lamc1. Whilst these additional laminin chains are expressed in the developing mouse neocortex, their levels may not be high enough to form a significant level of laminin 211 with overexpressed Lama2. Another point to consider is that overexpression of any laminin chain by electroporation is extremely difficult (and seldom performed) due to the size of an individual laminin chain; the coding sequences are 9.3 kilobases, 5.3 kilobases and 4.8 kilobases for Lama2, Lamb1 and Lamc1 respectively. In light of these considerations, we – like other groups in the field – decided to not overexpress the Lama2 α chain but rather add recombinant laminin 211 trimers to study their function.

2) The proliferating cells in the SVZ of ferret and human express Sox9; however, they also coexpress other transcription factors, e.g. Sox2 (as also shown by the authors), typical of proliferating progenitors in general (in fact, both are also expressed in the VZ). So, one may argue that Sox2 (or another coexpressed Sox factor, or other factor) could also be (together with Sox9) a candidate for these functions of the SVZ cells, as part of a genetic program of proliferating cells which has been recruited to SVZ cells through evolution. These possibilities should be discussed more in depth. Also, the effects (or not) of the Sox9 knockouts on cell proliferation should be mentioned. Overall, although the observed effects of ectopic Sox9 expression are very interesting, they are not in my opinion sufficient to qualify Sox9 as a "master regulator to promote neocortical expansion" (Abstract) nor to state (Discussion) that "The present study establishes Sox9 as a master regulator of BP proliferation and cell fate"; I would moderate these statements a little.

The scenario outlined by the reviewer is valid and now discussed in the revised manuscript.

Importantly, the new data showing reduced BP proliferation upon CRISPR/Cas9-mediated knockout of Sox9 in embryonic ferret neocortex substantially support the proposed role of Sox9 as a key regulator of BP proliferation. Nonetheless, we have moderated our statements as appropriate and deleted the term "master regulator".

[Editors' note: further revisions were suggested prior to acceptance, as described below.]

The reviewers have discussed the reviews with one another. While the paper is much improved, there are remaining concerns, detailed below that need to be addressed before a final decision can be made.Reviewer #1:The manuscript has been very nicely revised following the comments and concerns on its initial version. Unfortunately, there are still a handful of points that were not addressed properly:Point 4:It is very nice that the authors performed this analysis. However, it is very surprising and strange that in control electroporated cortex of mouse embryos more than 70% of RFP+pVim+ cells in SVZ-IZ are Pax6+ (less than 30% are Pax6-; new Figure 4—figure supplement 3). A large body of literature, including studies from the Huttner lab, demonstrates that the vast majority of mouse cortical BPs are bIPs, hence most mitotic cells in SVZ-IZ are Pax6-.

The reviewer is correct in stating that the "vast majority of mouse cortical BPs are bIPs". However, while the expression of Pax6 is clearly down-regulated concomitant with the transition from apical radial glia to bIPs, the Pax6 level in bIPs is not exactly zero. Rather, low Pax6 levels have been detected in mouse bIPs, and the proportion of mouse bIPs with low Pax6 levels increases during the course of neurogenesis. Thus, Fish et al., 2008 reported that ≈30% of E12.5-13.5 mouse BPs are weakly Pax6-positive. Hutton and Pevny 2011 (Developmental Biology 352, 40–47) stated for E16.5 mouse dorsal telencephalon "These data demonstrate that both SOX2 and PAX6 are maintained at high levels in RGCs, but at lower levels in TBR2-positive IPCs". An additional issue here is the sensitivity of Pax6 detection, which likely shows variability depending on the method used.

The reviewer had previously requested an analysis – using the marker Pax6 – of the effects of conditional Sox9 expression specifically on bIPs, which should be Pax6-negative. However, as some bIPs are weakly Pax6-positive (see above) and mouse basal radial glia in the subventricular zone are known to be Pax6-positive, we focused our analysis at E15.5 (Figure 4—figure supplement 3) on those BPs that were truly Pax6-negative, in order to be sure to quantify only bIPs. This issue has now been clarified in the re-revised manuscript, with addition of the above-mentioned Hutton and Pevny reference.

Point 9:This is incorrect. Multiple comparisons tests must be performed (i.e. ANOVA) when comparing value distributions, and then only if this reveals a statistical difference one must perform t-tests bin by bin.

We agree with the reviewer. As requested by the reviewer, we have now also performed a one-way ANOVA test, which revealed a statistical difference. We have added this information to the legend of Figure 8.

Point 11:The argument of this reviewer is well understood, but the issue here is that of expression levels: direct overexpression of laminin 211 increases BP proliferation, similar to overexpression of Sox9. But is the increased expression level of Laminin α2 caused by Sox9 overexpression of sufficient magnitude to really mediate this effect? In other words, the authors' conclusions would if they could rescue the effects of Sox9 overexpression by blockade of its effect on Laminin 211 expression levels.

As to this point raised by the reviewer, we would like to emphasize that we do not mean to imply that the increased expression level of laminin α2 mRNA caused by conditional Sox9 expression is necessarily of sufficient magnitude to alone mediate the increase in BP proliferation by laminin 211. Rather, we would like to stress that laminin α2 is not the only target of conditional Sox9 expression, and that laminin 211 was primarily used as a proof-of-principle experiment to support the concept: conditional Sox9 expression –> increased expression of ECM components –> BP proliferation. We have now clarified this in the re-revised manuscript.

Although we do not think that we should discuss this in the re-revised manuscript, it should be noted that a modest increase in laminin α2 could have a wider reaching effect than anticipated, as once it has formed a stable extracellular trimer, it can interact with multiple neighbouring cells for some time. In other words, a modest increase in laminin α2 expression does not automatically imply an only minor functional effect.

Regarding a possible reversal of the effect of conditional Sox9 expression on BP proliferation by blockade of its effect on laminin 211 expression, this would be an extremely difficult experiment to perform properly, as it would require to perform such a blockade selectively in the Tis21 lineage (please see Figure 4A, B). As an (admittedly) crude alternative, we have explored a blockade of laminin 211 function by using an integrin-blocking antibody, as the increase in BP proliferation by laminin 211 is mediated via integrin signaling. Specifically, we have exposed control- and Sox9-electroporated mouse neocortices to an integrin-blocking antibody for 24 hours in HERO culture, and analyzed proliferation in the SVZ by immunostaining for PCNA (n=1). As shown in Author response image 1, this experiment suggested that the increase in proliferation in the SVZ of Sox9-electroporated neocortex can be partially reversed by blocking ECM-induced integrin signaling using an integrin-blocking antibody. The limitation of this experiment is that integrins have other ECM ligands in addition to laminin 211, and so the integrin-blocking antibody would most likely also affect integrin signaling induced by ECM components other than laminin 211. Because of this ambiguity, we have not included this data in the re-revised manuscript.

**Author response image 1. respfig1:** Heterozygous *Tis21*-CreER^T2^ mouse embryos received tamoxifen at E12.5 and E13.5 and were subjected to *in utero* electroporation of the neocortex at E13.5 with either control construct or conditional Sox9 expression construct. Embryos were harvested 24 hours later, and neocortices were incubated with or without an integrin β1 blocking antibody in HERO culture for 24 hours, followed by immunostaining for PCNA. Quantification of the number of PCNA-positive, RFP-negative nuclei in the SVZ plus IZ per microscopic field of 150-µm apical width. White column, control construct, no integrin-blocking antibody; red column, conditional Sox9 expression construct, no integrin-blocking antibody; red checkered column, conditional Sox9 expression construct plus integrin-blocking antibody (Ab).

Point 12:The argument about background is acceptable, and the new high magnifications are nice, but the comment that apical mitoses are only partly visible was completely ignored. Although there is no change in apical mitosis abundance, these histological results must still be shown and demonstrated properly. The current format (same as in the previous manuscript version) is unacceptable.

In addition to the high magnification images showing basal mitoses that we had added to the revised Figure 9 as a new panel previously (now panel E''), we have now added another new panel to the re-revised Figure 9 (new panel E’) with high magnification images that clearly demonstrate apical mitoses, as requested by the reviewer.

Reviewer #3:The paper has been improved by the new experiments of Sox9 CRISPR-Cas9-mediated Sox9 KO in the ferret, which are very interesting. However, I maintain a major concern about the overemphasis given to Sox9 as a "key regulator" of cortical size increase during evolution, which is not, in my opinion, supported by the data (see below). The findings remain interesting, but overemphasis in interpretation is misleading, and should be avoided.

In line with the concern raised by the reviewer, we have completely eliminated the term "key regulator" from the manuscript and toned down our conclusions regarding the role of Sox9 in the evolutionary expansion of the neocortex.

Specifically:

The final sentence of the Abstract, which previously read:

“Our findings demonstrate that Sox9 exerts concerted effects on transcription, BP proliferation, neuron production, and neurogenic vs. gliogenic BP cell fate, suggesting that Sox9 acts as key regulator in the subventricular zone to promote neocortical expansion”, now reads:

“Our findings demonstrate that Sox9 exerts concerted effects on transcription, BP proliferation, neuron production, and neurogenic vs. gliogenic BP cell fate, suggesting that Sox9 may have contributed to promote neocortical expansion.”

The first sentence of the Discussion, which previously read:

“The present study establishes Sox9 as a key regulator of BP proliferation and cell fate”, now reads:

“The present study establishes Sox9 as an important regulator of BP proliferation and cell fate.”

The final sentence of the Discussion, which previously read:

“Hence, when collectively considering the effects of Sox9 on transcription, BP proliferation and fate, and neuron production, this transcription factor shows all the hallmarks expected for a key regulator that acts in the SVZ to promote neocortical expansion”, now reads:

“Hence, when collectively considering its effects on transcription, BP proliferation and fate, and neuron production, Sox9 acting in the SVZ shows many of the hallmarks expected for a transcription factor contributing to promote neocortical expansion.”

Sox9 activation in mouse BP leads to a statistically significant, but very moderate in absolute terms, increase of BP proliferation, starting from a very low background of spontaneous (i.e. not induced by transgene) BP proliferation. The authors should take this point into serious consideration in the Discussion and throughout the paper, and substantially attenuate their proposal that Sox9 expression in BP is a key element explaining the expansion of the cortex in evolution.

As requested by the reviewer and as specified above, we have substantially attenuated our previous proposal that Sox9 expression in BPs is a key element explaining the expansion of the neocortex in evolution.

The important effects shown by the KO in the ferret (though see point 2 below, to be clarified), which are certainly a worthy addition to the paper, should be reconciled with the relative ineffectiveness of Sox9 overexpression in mouse; perhaps Sox9 expression in BP in the ferret is accompanied by expression of other important factors, and it is the synergy of Sox9 with these other factors that matters in the expansion of the cortex.

We would like to thank the reviewer for raising this excellent possibility. In considering this possibility, we noticed that Sox8, with which Sox9 can heterodimerize, shows a similar expression pattern with regard to the neocortical germinal zones as Sox9, being expressed in embryonic mouse neocortex essentially only in the VZ but in fetal human neocortex in the VZ, ISVZ and OSVZ. Hence, if there is synergy of Sox9 with Sox8, such differential expression of the two transcription factors between species could explain why the effects of Sox9 KO in embryonic ferret neocortex appear to be stronger than those of conditional Sox9 expression in embryonic mouse neocortex. We have now added this scenario to the re-revised Discussion.

In the new Figure 3 (Sox9 ko in ferret), the PCNA -mCherry/mCherry -positive cells within the VZ are reported to be numerically similar between Sox9-mutant and control ferrets (histograms in Figure 3B), however the intensity of PCNA staining in the Sox9-mutant VZ (Figure 3A) is much lower than in control. This makes one fear that the 50% reduction of PCNA-positive cells reported for the mutant OSVZ (Figure 4B) may be contributed to by a lower efficiency of the staining. This should be clarified, the image shown and the data in the histogram should be consistent.

The possibility raised by the Reviewer does not really apply to our data, for the following reasons.

First of all, and importantly, the intensity of the PCNA immunostaining was not taken into consideration for the quantification shown in Figure 3B. Rather, the cells were classified either as PCNA+, even if the PCNA immunoreactivity was low, or PCNA–, if really no PCNA immunoreactivity could be detected. In other words, the difference in PCNA immunostaining intensity did not affect the quantification shown in Figure 3B, because we counted even the only weakly PCNA-positive cells as PCNA+.

Second, it should be noted that in the two PCNA immunostainings shown in Figure 3A, the brightest cells are equally bright for both the control and the Sox9 KO. This is obvious when comparing, between the control and the Sox9 KO, (i) the PCNA^+^ cells at the apical surface, and (ii) the brightest cells in the OSVZ. However, as noted by the reviewer, in the basal region of the VZ, which contains basal progenitors migrating to the SVZ and apical progenitors in S-phase, the staining of the PCNA^+^ cells is less intense in the Sox9 KO than the control. We believe that this actually is a phenotype of the Sox9 KO, in particular when considering that many of the cells in the basal VZ are newborn basal progenitors that are migrating away from the VZ. This decrease in PCNA immunostaining intensity would be in accordance with the main finding of this figure, which is that the Sox9 KO causes a reduction in cycling basal progenitors. Please note that this conclusion can be reached independently of the PCNA immunostaining when considering the abundance of mitotic progenitors in the ISVZ and OSVZ as revealed by phospho-vimentin immunostaining (Figure 3C).

We have now clarified this issue in the legend to Figure 3.